# An ensemble penalized regression method for multi-ancestry polygenic risk prediction

Jingning Zhang [1] ✉, Jianan Zhan[2], Jin Jin [3], Cheng Ma[4], Ruzhang Zhao[1], Jared O'Connell[2], Yunxuan Jiang[2], 23andMe Research Team*, Bertram L. Koelsch[2], Haoyu Zhang [5] & Nilanjan Chatterjee [1,6] ✉

Great efforts are being made to develop advanced polygenic risk scores (PRS) to improve the prediction of complex traits and diseases. However, most existing PRS are primarily trained on European ancestry populations, limiting their transferability to non-European populations. In this article, we propose a novel method for generating multi-ancestry Polygenic Risk scOres based on enSemble of PEnalized Regression models (PROSPER). PROSPER integrates genome-wide association studies (GWAS) summary statistics from diverse populations to develop ancestry-specific PRS with improved predictive power for minority populations. The method uses a combination of $\mathscr{L}_1$ (lasso) and $\mathscr{L}_2$ (ridge) penalty functions, a parsimonious specification of the penalty parameters across populations, and an ensemble step to combine PRS generated across different penalty parameters. We evaluate the performance of PROSPER and other existing methods on large-scale simulated and real datasets, including those from 23andMe Inc., the Global Lipids Genetics Consortium, and All of Us. Results show that PROSPER can substantially improve multi-ancestry polygenic prediction compared to alternative methods across a wide variety of genetic architectures. In real data analyses, for example, PROSPER increased out-of-sample prediction $R^2$ for continuous traits by an average of 70% compared to a state-of-the-art Bayesian method (PRS-CSx) in the African ancestry population. Further, PROSPER is computationally highly scalable for the analysis of large SNP contents and many diverse populations.

Tens of thousands of single nucleotide polymorphisms (SNP) have been mapped to human complex traits and diseases through genome-wide association studies (GWAS)[1,2]. Though each SNP only explains a small fraction of variation of the underlying phenotype, polygenic risk scores (PRS), which aggregate the genetic effects of many loci, can have a substantial ability to predict traits and stratify populations by underlying disease risks[3–12]. However, as existing GWAS to date have been primarily conducted in European ancestry populations (EUR)[13–16], recent studies have consistently shown that the transferability of EUR-derived PRS to non-EUR populations often is suboptimal and in particular poor for African Ancestry populations[17–21].

Despite growing efforts of conducting genetic research on minority populations[22–25], the gap in sample sizes between EUR and non-EUR populations is likely to persist in the foreseeable future. As the

[1]Department of Biostatistics, Johns Hopkins Bloomberg School of Public Health, Baltimore, MD, USA. [2]23andMe Inc., Sunnyvale, CA, USA. [3]Department of Biostatistics, Epidemiology, and Informatics, University of Pennsylvania, Philadelphia, PA, USA. [4]Department of Statistics, University of Michigan, Ann Arbor, MI, USA. [5]Division of Cancer Epidemiology and Genetics, National Cancer Institute, Bethesda, MD, USA. [6]Department of Oncology, School of Medicine, Johns Hopkins University, Baltimore, MD, USA.*A list of authors and their affiliations appears at the end of the paper. ✉e-mail: jingningzhang238@gmail.com; nilanjan@jhu.edu

performance of PRS largely depends on the sample size of training GWAS[3,26], using single-ancestry methods[27–31] to generate PRS for a minority population, using data from that population alone may not achieve ideal results. To address this issue, researchers have developed methods for generating powerful PRS by borrowing information across diverse ancestry populations[32]. For example, Weighted PRS[33] combines single-ancestry PRS generated from each population using weights that optimize performance for a target population. Bayesian methods have also been proposed that generate improved PRS for each population by jointly modeling the effect-size distribution across populations[34,35]. Recently, our group proposed a new method named CT-SLEB[21], which extends the clumping and thresholding (CT)[36] method to multi-ancestry settings. The method uses an empirical-Bayes (EB) approach to estimate effect sizes by borrowing information across populations and a super learning model to combine PRSs under different tuning parameters. However, the optimality of the methods depends on many factors, including the ability to account for heterogeneous linkage disequilibrium (LD) structure across populations and the adequacy of the models for underlying effect-size distribution[3,26]. In general, our extensive simulation studies and data analyses suggest that no method is uniformly the most powerful, and exploration of complementary methods will often be needed to derive the optimal PRS in any given setting[21].

In this article, we propose a novel method for generating multi-ancestry Polygenic Risk scOres based on an enSemble PEnalized Regression (PROSPER) using GWAS summary statistics and validation datasets across diverse populations. The method incorporates $\mathcal{L}_1$ penalty functions for regularizing SNP effect sizes within each population, an $\mathcal{L}_2$ penalty function for borrowing information across populations, and a flexible but parsimonious specification of the underlying penalty parameters to reduce computational time. Further, instead of selecting a single optimal set of tuning parameters, the method combines PRS generated across different populations and tuning parameters using a final ensemble regression step. We compare the predictive performance of PROSPER with a wide variety of single- and multi-ancestry methods using simulation datasets from our recent study[21] across five populations (EUR, African (AFR), Ad Mixed American (AMR), East Asian (EAS), and South Asian (SAS))[21]. Furthermore, we evaluate these methods using a variety of real datasets from 23andMe Inc. (23andMe), the Global Lipids Genetics Consortium (GLGC)[37], All of Us (AoU)[38], and the UK Biobank study (UKBB)[39]. Results from these analyses indicate that PROSPER is a highly promising method for generating the most powerful multi-ancestry PRS across diverse types of complex traits. Computationally, PROSPER is also exceptionally scalable compared to other advanced methods.

## Results

### Method overview

PRSOSPER is a method designed to improve prediction performance for PRS across distinct ancestral populations by borrowing information across ancestries (Fig. 1). It can integrate large EUR GWAS with smaller GWAS from non-EUR populations. Ideally, individual-level tuning data are needed for all populations, because the method needs optimal parameters from single-ancestry analysis as an input; however, even when data is only available for a target population, PRSOSPER can still be performed, and the PRS will be optimized and validated toward the target population. The method can account for population-specific genetic variants, allele frequencies, and LD patterns and use computational techniques for penalized regressions for fast implementation.

### PROSPER

Assuming a continuous trait, we first consider a standard linear regression model for underlying individual-level data for describing the relationship between trait values and genome-wide genetic variants across $M$ distinct populations. Let $\mathbf{Y}_i$ denote the $n_i \times 1$ vector of

trait values, $\mathbf{X}_i$ denote the $n_i \times p_i$ genotype matrix, $\boldsymbol{\beta}_i$ denote the $p_i \times 1$ vector of SNP effects, and $\boldsymbol{\epsilon}_i$ denote the $n_i \times 1$ vector of random errors for the $i^{\text{th}}$ population. We assume underlying linear regression models of the form $\mathbf{Y}_i = \mathbf{X}_i \boldsymbol{\beta}_i + \boldsymbol{\epsilon}_i, i = 1, \ldots, M$; and intend to solve the linear regression system by least square with a combination of $\mathcal{L}_1$ (lasso)[40] and $\mathcal{L}_2$ (ridge)[41] penalties in the form

$$\sum_{1 \le i \le M} \frac{1}{n_i} (\mathbf{Y}_i - \mathbf{X}_i \boldsymbol{\beta}_i)^T (\mathbf{Y}_i - \mathbf{X}_i \boldsymbol{\beta}_i)$$
$$+ \sum_{1 \le i \le M} 2\lambda_i \|\boldsymbol{\beta}_i\|_1^1$$
$$+ \sum_{1 \le i_1 < i_2 \le M} c_{i_1 i_2} \|\boldsymbol{\beta}_{i_1}^{s_{i_1 i_2}} - \boldsymbol{\beta}_{i_2}^{s_{i_1 i_2}}\|_2^2$$

where $\lambda_i, i = 1, \ldots, M$ are the population-specific tuning parameters associated with the lasso penalty; $\boldsymbol{\beta}_{i_1}^{s_{i_1 i_2}}$ and $\boldsymbol{\beta}_{i_2}^{s_{i_1 i_2}}$ denote the vectors of effect-sizes for SNPs for the $i_1$-th and $i_2$-th populations, respectively, restricted to the set of shared SNPs ($s_{i_1 i_2}$) across the pair of the populations; and $c_{i_1 i_2}, 1 \le i_1 < i_2 \le M$ are the tuning parameters associated with the ridge penalty imposing effect-size similarity across pairs of populations.

In the above, the first part, $\sum_{1 \le i \le M} 2\lambda_i \|\boldsymbol{\beta}_i\|_1^1$, uses a lasso penalty. Lasso can produce sparse solution[40] and recent PRS studies that have implemented the lasso penalty in the single-ancestry setting have shown its promising performance[28,29]. The second part, $\sum_{1 \le i_1 < i_2 \le M} c_{i_1 i_2} \|\boldsymbol{\beta}_{i_1}^{s_{i_1 i_2}} - \boldsymbol{\beta}_{i_2}^{s_{i_1 i_2}}\|_2^2$, uses a ridge penalty. As it has been widely shown that the causal effect sizes of SNPs tend to be correlated across populations[42,43], we propose to use the ridge penalty to induce genetic similarity across populations. Compared to the fused lasso[44], which uses lasso penalty for the differences, we use ridge penalty instead, which allows a small difference in SNP effects across populations rather than truncating them to zero. The solutions for population-specific effect size using the combined lasso and ridge penalties can be sparse.

The estimate of $\boldsymbol{\beta}_i, i = 1, \ldots, M$ in the above individual-level linear regression systems can be obtained by minimizing the above least square objective function. Following the derivation of lassosum[28], a single-ancestry method for fitting the lasso model to GWAS summary statistics data, we show that the objective function for individual-level data can be approximated using GWAS summary statistics and LD reference matrices by substituting $\frac{1}{n_i} \mathbf{X}_i^T \mathbf{X}_i$ by $\mathbf{R}_i$, where $\mathbf{R}_i$ is the estimated LD matrix based on a reference sample from the $i$-th population, and $\frac{1}{n_i} \mathbf{X}_i^T \mathbf{y}_i$, by $\mathbf{r}_i$, where $\mathbf{r}_i$ is the GWAS summary statistics in the $i$-th population. Therefore, the objective function of the summary-level model can be written as

$$\sum_{1 \le i \le M} (\boldsymbol{\beta}_i^T (\mathbf{R}_i + \delta_i \mathbf{I}) \boldsymbol{\beta}_i - 2\boldsymbol{\beta}_i^T \mathbf{r}_i + 2\lambda_i \|\boldsymbol{\beta}_i\|_1^1) + \sum_{1 \le i_1 < i_2 \le M} c_{i_1 i_2} \|\boldsymbol{\beta}_{i_1}^{s_{i_1 i_2}} - \boldsymbol{\beta}_{i_2}^{s_{i_1 i_2}}\|_2^2$$

where the additional tuning parameters $\delta_i, i = 1, \ldots, M$, are introduced for regularization of the LD matrices across the different populations[29]. For a fixed set of tuning parameters, the above objective function can be solved using fast coordinate descent algorithms[45] by iteratively updating each element of $\boldsymbol{\beta}_i, i = 1, \ldots, M$ (see "Obtain PROSPER solution" under "Methods").

### Reducing tuning parameters

For the selection of tuning parameters, we assume we have access to individual-level data across the different populations which are independent of underlying GWAS from which summary statistics are generated. The above setting involves three sets of tuning parameters, $\{\delta_i\}_{i=1}^M$, $\{\lambda_i\}_{i=1}^M$, and $\{c_{i_1 i_2}\}_{1 \le i_1 < i_2 \le M}$, totaling to the number of $M + M + \frac{M(M-1)}{2}$. As grid search across many combinations of tuning parameter values can be computationally intensive, we propose to

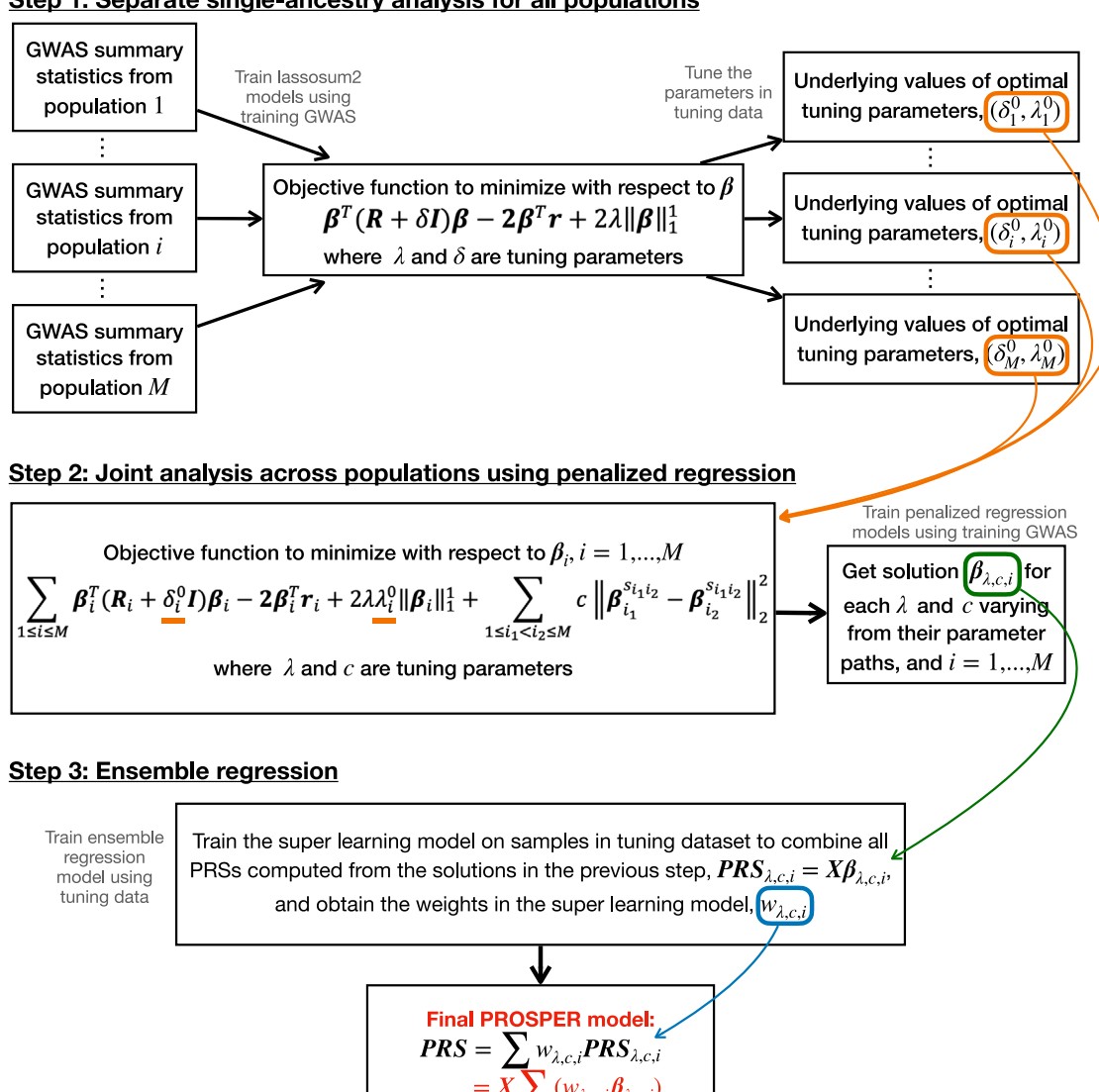

**Fig. 1 | Detailed flowchart of PROSPER.** The analysis of $M$ populations in PROSPER involves three key steps: (1) Separate single-ancestry analysis for all populations $i = 1, \ldots, M$; (2) Joint analysis across populations using penalized regression; (3) Ensemble regression. In step 1, the training GWAS data is used to train lassosum2 models, and the tuning data is used to obtain the optimal tuning parameters in a single-ancestry analysis. In step 2, the training GWAS and the optimal tuning parameter values from step 1 are used to train the joint cross-population penalized regression model, and obtain solution $\boldsymbol{\beta}_{\lambda,c,i}$ for each $\lambda$ and $c$. In step 3, the tuning data is used to train the super learning model for the ensemble of PRSs computed from the solutions in step 2, $\textbf{PRS}_{\lambda,c,i} = \mathbf{X}\boldsymbol{\beta}_{\lambda,c,i}$. The final PRS is computed as $\textbf{PRS} = \mathbf{X}(\sum \boldsymbol{w}_{\lambda,c,i}\boldsymbol{\beta}_{\lambda,c,i})$, where $w_{\lambda,c,i}$ are the weights from the super learning model. Refer to the "Method Overview" section in the main text for a full explanation of all notations in the flowchart.

reduce the search range by a series of steps. First, we use lassosum2[29] to analyze GWAS summary statistics and tuning data from each ancestry population by itself and obtain underlying values of optimal tuning parameters, $(\delta_i^0, \lambda_i^0)$ for $i = 1, \ldots, M$; if tuning data is only available for the target population, the $(\delta_i^0, \lambda_i^0)$ for other populations can be optimized towards the target population. For fitting PROSPER, we fix $\delta_i = \delta_i^0$ for $i = 1, \ldots, M$ as these are essentially used to regularize estimates of population-specific LD matrices. We note that the optimal $\{\lambda_i\}_{i=1}^M$ depend on sample sizes of underlying training GWAS (Supplementary Fig. 1), and thus should not be arbitrarily assumed to be equal across all populations. Considering that the optimal tuning parameters associated with the $\mathscr{L}_1$ penalty function from the single-ancestry analyses should reflect the characteristics of GWAS data, which includes underlying sparsity of effect sizes and sample sizes, we propose to specify the $\mathscr{L}_1$-tuning parameters in PROSPER as

$\lambda_i = \lambda\lambda_i^0$, i.e., they are determined by the corresponding tuning parameters from the ancestry-specific analysis except for the constant multiplicative factor $\lambda$. Finally, for computational feasibility, we further assume that effect sizes across all pairs of populations have a similar degree of homogeneity and thus set all $\{c_{i_1i_2}\}_{1 \leq i_1 < i_2 \leq M}$ to be equal to $c$. We will later discuss this assumption and perform a sensitivity analysis (see Discussion). By using the above assumptions, the objective function to minimize with respect to $\boldsymbol{\beta}_i, i = 1, \ldots, M$, becomes

$$\sum_{1 \leq i \leq M} (\boldsymbol{\beta}_i^T(\mathbf{R}_i + \delta_i^0\mathbf{I})\boldsymbol{\beta}_i - 2\boldsymbol{\beta}_i^T\mathbf{r}_i + 2\lambda\lambda_i^0\|\boldsymbol{\beta}_i\|_1^1) + \sum_{1 \leq i_1 < i_2 \leq M} c\|\boldsymbol{\beta}_{i_1}^{s_{i_1i_2}} - \boldsymbol{\beta}_{i_2}^{s_{i_1i_2}}\|_2^2$$

where $\lambda$ and $c$ are the only two tuning parameters needed for lasso penalty and genetic similarity penalty, respectively.

## Ensemble

Using an ensemble method to combine PRS has been shown to be promising in CT-type methods as opposed to picking an optimal threshold[21,36]. In general, a specific form of the penalty function, or equivalently a model for prior distribution in the Bayesian framework, may not be able to adequately capture the complex nature of the underlying distribution of the SNPs across diverse populations. We conjecture that when effect size distribution is likely to be mis-specified, an ensemble method, which combines PRS across different values of tuning parameters instead of choosing one optimal set, is likely to improve prediction. Therefore, as a last step, we obtain the final PROSPER model using an ensemble method, super learning[46–48], implemented in the *SuperLearner* R package, to combine PRS generated from various tuning parameter settings and optimized using tuning data from the target population. The super learner we use here was based on three supervised learning algorithms, including lasso[40], ridge[41], and linear regression (see "Methods").

## Results

### Methods comparison on simulated data

We conducted simulation analyses on continuous traits under various genetic architectures[21] to evaluate the performance of different methods that can be categorized into five groups: single-ancestry methods trained from target GWAS data (single-ancestry method), single-ancestry methods trained from EUR GWAS data (EUR PRS-based method), simple multi-ancestry methods by weighting single-ancestry PRS (weighted PRS), recently published multi-ancestry methods (existing multi-ancestry methods), and our proposed method PROSPER. Single-ancestry methods include CT[36], LDpred2[30], and lassosum2[29]. Existing multi-ancestry methods include PRS-CSx[34] and CT-SLEB[21]. All of the methods were implemented using the latest available version of the underlying software. The performance of the methods is evaluated by $R^2$ measured on validation samples independent of training and tuning datasets. Analyses in this and the following sections are restricted to a total of 2,586,434 SNPs, which are included in either HapMap 3 (HM3)[49] or the Multi-Ethnic Genotyping Arrays (MEGA) chips array[50]. LD reference samples for all five ancestries, EUR, AFR, AMR, EAS, and SAS, in this and the following sections, are from 1000 Genomes Project (Phase 3)[51] (1000G).

The results (Fig. 2, Supplementary Figs. 2–5, and Supplementary Data 1–5) show that multi-ancestry methods generally exhibit superior performance compared to single-ancestry methods. Weighted PRS generated from methods modeling LD (LDpred2 and lassosum2) can lead to a noticeable improvement in performance (green bars in Fig. 2). Notably, PROSPER shows robust performance uniformly across different scenarios. When the sample size of the target non-EUR population is small ($N_{target}$ = 15K) (Fig. 2a), PROSPER has comparable good performance with other multi-ancestry methods, such as weighted LDpred2 and PRS-CSx, under a high degree of polygenicity ($p_{causal}$ = 0.01). However, under the same sample size setting and lower polygenicity ($p_{causal}$ = 0.001 and $5 \times 10^{-4}$), PRS-CSx and CT-SLEB outperform PROSPER, with the margin of improvement increasing as the strength of negative selection decreases (strong negative selection in Fig. 2a, mild negative selection in Supplementary Fig. 2a, and no negative selection in Supplementary Fig. 3a). When the sample size of the target population is large ($N_{target}$ = 80K) (Fig. 2b and Supplementary Figs. 2–5b), PROSPER almost uniformly outperforms all other methods, particularly for the AFR population, and weighted LDpred2 remains a close competitor.

We further compare the computational efficiency of PROSPER in comparison to PRS-CSx, the state-of-the-art Bayesian method available for generating multi-ancestry PRS. We train PRS models for the two methods using simulated data for chromosome 22 using a single core with AMD EPYC 7702 64-Core Processors running at 2.0 GHz. We observe (Supplementary Data 6) that PROSPER is 37 times faster than PRS-CSx (3.0 vs. 111.1 minutes) in a two-ancestry analysis including AFR and EUR; and 88 times faster (6.8 vs. 595.8 minutes) in the analysis of all five ancestries. The memory usage for PRS-CSx is about 2.8 times smaller than PROSPER (0.78 vs. 2.24 Gb in two-ancestry analysis, and 0.84 vs. 2.35 Gb in five-ancestry analysis).

### 23andMe data analysis

We applied various methods to GWAS summary statistics available from the 23andMe, Inc. to predict two continuous traits, heart metabolic disease burden and height; as well as five binary traits, any cardiovascular disease (any CVD), depression, migraine diagnosis, morning person, and sing back musical note (SBMN). The datasets are available for all five ancestries, African American (AA), Latino, EAS, EUR, and SAS. The methods are tuned and validated on a set of independent individuals of the corresponding ancestry from the 23andMe participant cohort (see the section of "Real data analysis" under "Methods" for data description, and Supplementary Data 7 and 8 for sample sizes used in training, tuning and validation). In an earlier version of the analysis, we had analyzed the data using an older version LDpred2 in its package of bigsnpr (version 1.8) that was available when the project was initiated. Quality control analysis following comments from one of the reviewers indicated problem with convergence of those results. As we were not able to further update the analysis using the most recent version of the LDpred2 in its package of bigsnpr (version 1.12) due to time constraint of the 23andMe team, we did not report results from LDpred2 and its corresponding EUR and weighted methods in this section of 23andMe data analysis.

From the analysis of two continuous traits (Fig. 3 and Supplementary Data 9), we observe that lassosum2 and its related methods (EUR lassosum2 and weighted lassosum2) generally perform better than CT and its related methods. On the basis of the advantage of lassosum2, PROSPER further improves the performance, and for most of the settings, outperforms all alternative methods, including PRS-CSx and CT-SLEB. PROSPER demonstrates particularly remarkable improvement for both traits in AA and Latino (26.9% relative improvement in $R^2$ over the second-best method on average, yellow cells in Supplementary Data 10) (first two panels in Fig. 3a, b). For EAS and SAS, PROSPER is slightly better than other methods, except for heart metabolic disease burden of SAS (the last panel in Fig. 3a), which has the smallest sample size (~20 K).

The results from the analysis of the binary traits (Fig. 4 and Supplementary Data 9) show that PROSPER generally exhibits better performance (7.8% and 12.3% relative improvement in logit-scale variance (see "Methods") over CT-SLEB and PRS-CSx, respectively, averaged across populations and traits) (blue and red cells, respectively, in Supplementary Data 10). A similar trend is observed for the analyses of AA and Latino, where PROSPER usually has the best performance (first two panels in Fig. 4a–e). In general, no single method can uniformly outperform others. Weighted lassosum2 has outstanding performance for depression (Fig. 4b), while PROSPER is superior for morning person (Fig. 4d). PRS-CSx shows a slight improvement in the analysis of migraine diagnosis for EAS populations (last second panel in Fig. 4c), and CT-SLEB performs the best in the analysis of any CVD for SAS population (last panel in Fig. 4a).

### GLGC and AoU data analysis

Considering the uncommonly huge sample sizes from 23andMe, we further applied alternative methods for the analysis of two other real datasets, GLGC and AoU. The GWAS summary statistics from GLGC for four blood lipid traits, high-density lipoprotein (HDL), low-density lipoprotein (LDL), log-transformed triglycerides (logTG), and total cholesterol (TC), are publicly downloadable and available for all five ancestries, African/Admixed African, Hispanic, EAS, EUR, and SAS (see "Methods"' for data description, and Supplementary Data 7 for sample sizes). Further, we generated GWAS summary statistics data from the

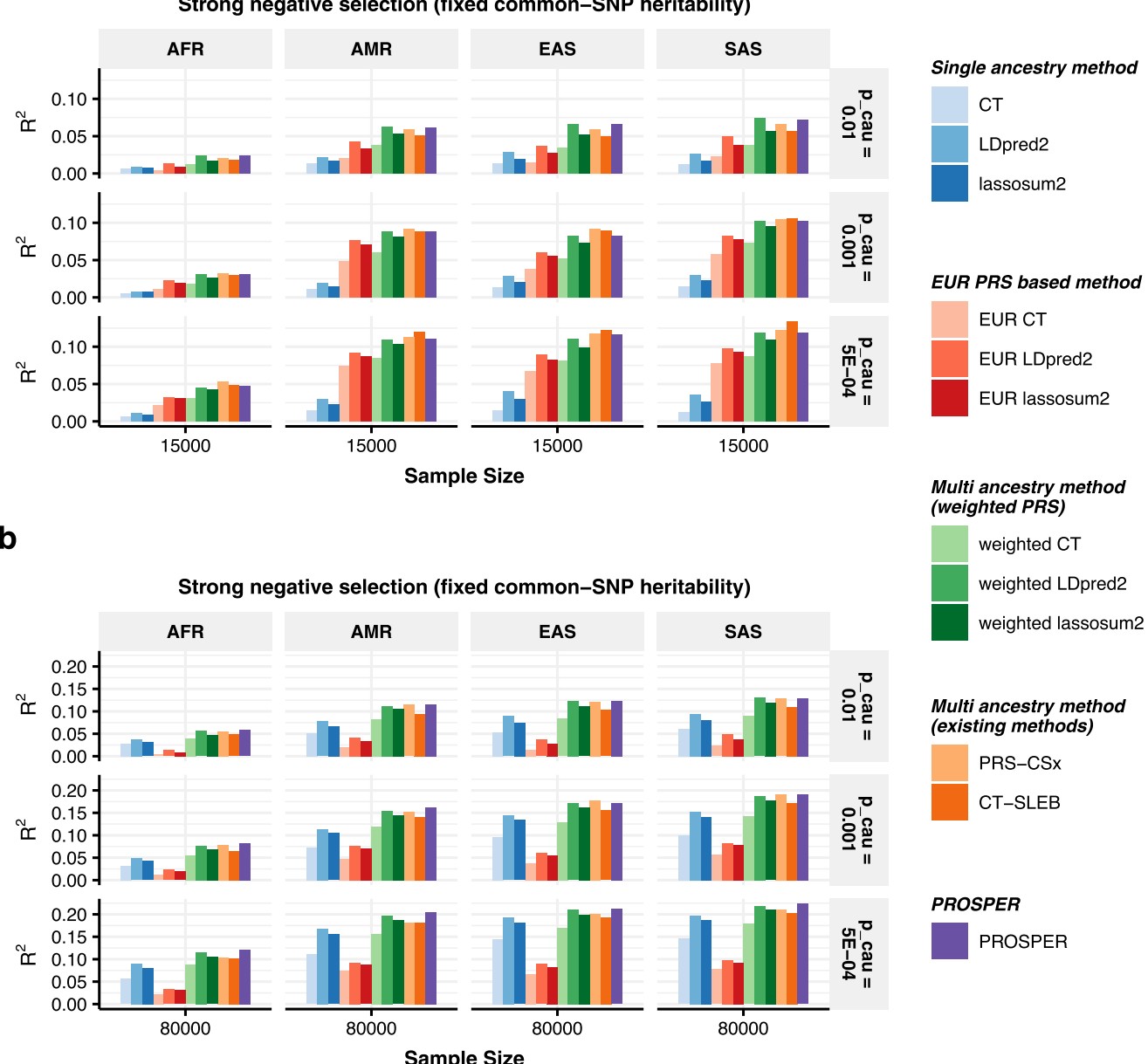

**Fig. 2 | Performance comparison of alternative methods on simulated data generated with different sample sizes and genetic architectures under strong negative selection and fixed common-SNP heritability.** Data are simulated for continuous phenotype under a strong negative selection model and three different degrees of polygenicity (top panel: $p_{causal}$ = 0.01, middle panel: $p_{causal}$ = 0.001, and bottom panel: $p_{causal}$ = $5 \times 10^{-4}$). Common SNP heritability is fixed at 0.4 across all populations, and the correlations in effect sizes for share SNPs between all pairs of populations is fixed at 0.8. The sample sizes for GWAS training data are assumed to be **a** $n$ = 15,000, and **b** $n$ = 80,000 for the four non-EUR target populations; and is fixed at $n$ = 100,000 for the EUR population. PRS generated from all methods are tuned in $n$ = 10,000 samples, and then tested in $n$ = 10,000 independent samples in each target population. The PRS-CSx package is restricted to SNPs from HM3, whereas other alternative methods use SNPs from either HM3 or MEGA. Bars in the figure show the performance of $R^2$ for each method in each dataset. Colors are described on the right side of the figure. Source data are provided in Supplementary Data 1.

AoU study for two anthropometric traits, body mass index (BMI) and height, for individuals from three ancestries, AFR, EUR, and Latino/Admixed American (see "Methods" for data description, and Supplementary Data 7 for sample sizes). Both the blood lipid traits and anthropometric traits have corresponding phenotype data available in the UKBB, which we use to perform tuning and validation (see "Real data analysis" under "Methods" for the ancestry composition, and Supplementary Data 8 for sample sizes). Given the limited sample sizes of genetically inferred AMR ancestry individuals in UKBB, we do not

report the performance of PRS on AMR individuals in UKBB. In these analyses, we implemented LDpred2 method using the latest version of the software (version 1.12).

Results from analysis of four blood lipid traits (Fig. 5 and Supplementary Data 11) from GLGC and UKBB show that weighted PRS methods substantially outperform alternative methods. In particular, we observe that the weighted lassosum2 outperforms the other two weighted methods. Furthermore, our proposed method, PROSPER, shows improvement over weighted lassosum2 in both AFR and SAS

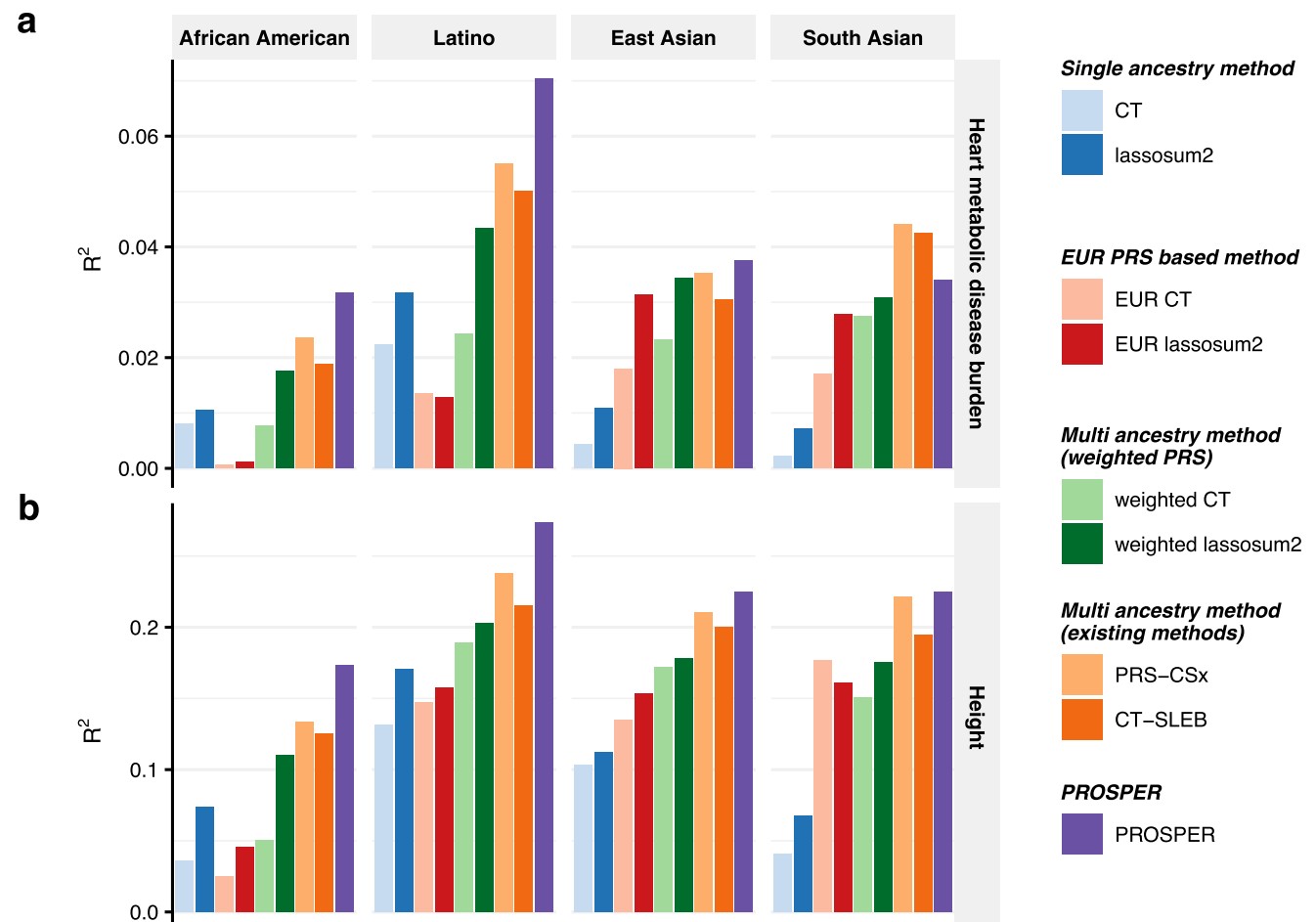

**Fig. 3 | Performance comparison of alternative methods for prediction of two continuous traits in 23andMe.** We analyzed two continuous traits, **a** heart metabolic disease burden and **b** height. PRS are trained using 23andMe data that available for five populations: African American, Latino, EAS, EUR, and SAS, and then tuned in an independent set of individuals from 23andMe of the corresponding ancestry. Performance is reported based on adjusted $R^2$ accounting for sex, age and PC1-5 in a held-out validation sample of individuals from 23andMe of the corresponding ancestry. The ratio of sample sizes for training, tuning and validation is roughly about 7:2:1, and detailed numbers are in Supplementary Data 7 and 8. The PRS-CSx package is restricted to SNPs from HM3, whereas other alternative methods use SNPs from either HM3 or MEGA. LDpred2 and its corresponding EUR and weighted methods are excluded to avoid misinterpretation, as a result of our collaboration restrictions with 23andMe, Inc., preventing us from updating these methods to the latest version of its package. Bars in the figure show the performance of adjusted $R^2$ for each method in each dataset. Colors are described on the right side of the figure. Source data are provided in Supplementary Data 9.

(13.5% and 12.3% relative improvement in $R^2$, respectively, averaged across traits) (green and orange cells, respectively, in Supplementary Data 12), but not in EAS. Notably, PROSPER outperforms PRS-CSx and CT-SLEB in most scenarios (34.2% and 37.7% relative improvement in $R^2$, respectively, averaged across traits and ancestries) (blue and red cells, respectively, in Supplementary Data 12), with the improvement being particularly remarkable for the AFR population (Fig. 5) in which PRS development tends to be the most challenging.

The results from AoU and UKBB (Fig. 6 and Supplementary Data 13) show that PROSPER generates the most predictive PRS for the two analyzed anthropometric traits for the AFR population. It appears that Bayesian and penalized regression methods that explicitly model LD tend to outperform corresponding CT-type methods (CT, EUR CT, and weighted CT) which excluded correlated SNPs. Among weighted methods, both LDpred2 and lassosum2 show major improvement over the corresponding CT method. Further, for both traits, PROSPER shows remarkable improvement over the best of the weighted methods and the two other advanced methods, PRS-CSx and CT-SLEB (91.3% and 76.5% relative improvement in $R^2$, respectively, averaged across the two traits) (blue and red cells, respectively, in Supplementary Data 14).

### Gain from PROSPER over lassosum2

To investigate whether the additional gain from PROSPER arises from modeling shared effects across populations or from combining PRS with super learning, we further employ a super learning step for lassosum2 (termed as advanced weighted lassosum2) as a point of comparison. The results in simulations (Supplementary Figs. 6–10 and Supplementary Data 15) indicate that PROSPER consistently has more advantage than the advanced weighted lassosum2 in all scenarios. The results in real data (Supplementary Figs. 11 and 12 and Supplementary Data 16) show that the performance of the two methods depends on traits and ancestries. PROSPER has comparable performance with advanced weighted lassosum2 in AFR; while PROSPER outperforms advanced weighted lassosum2 almost in all scenarios in SAS and EAS. In summary, PROSPER has 41.1% relative improvement in $R^2$ over advanced weighted lassosum2 on average across all ancestries and all traits in GLGC and AoU. We were not able to perform this analysis in 23andMe due to time constraint of the 23andMe team.

### Discussion

In this article, we propose PROSPER as a powerful method that can jointly model GWAS summary statistics from multiple ancestries by an

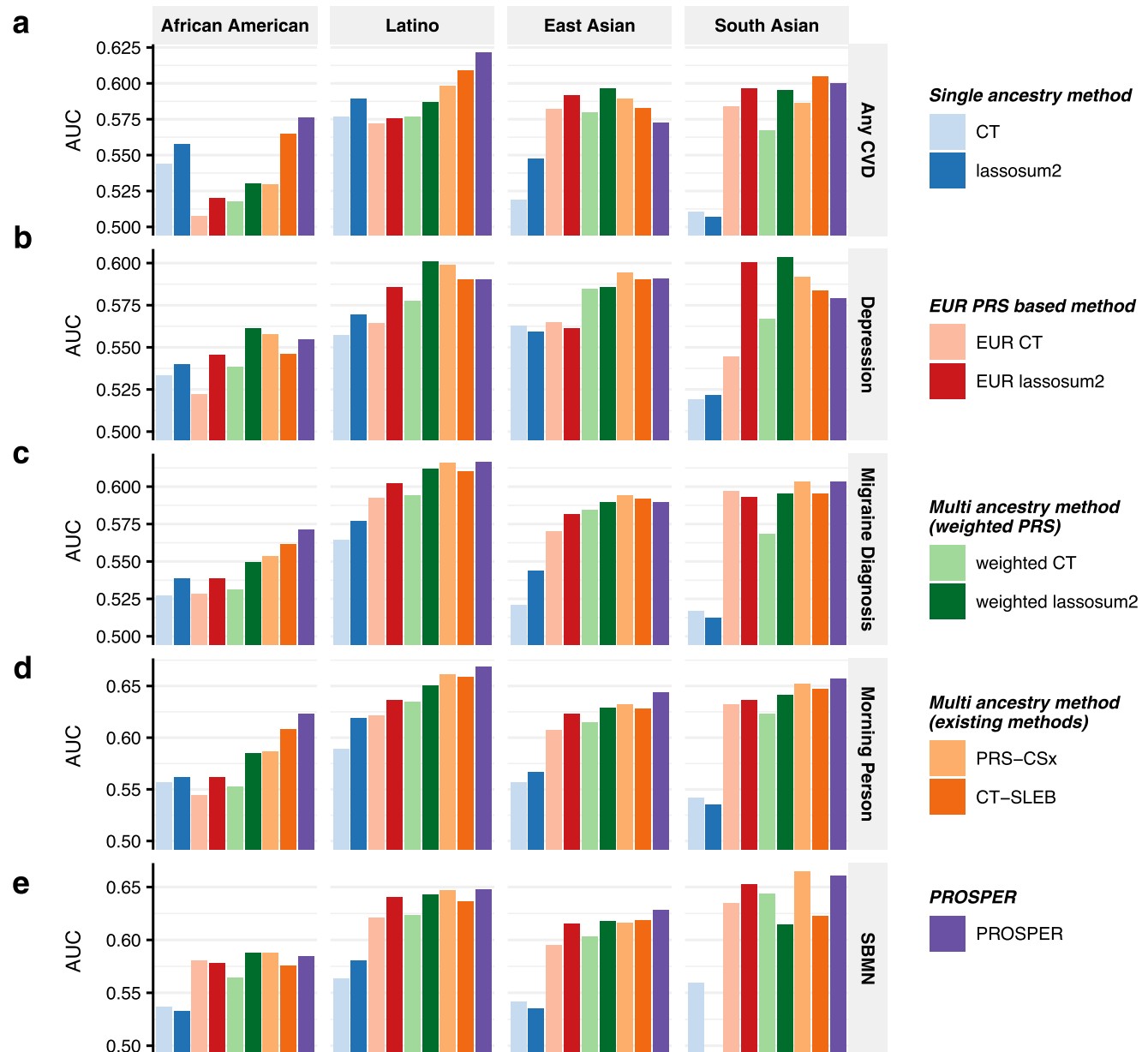

**Fig. 4 | Performance comparison of alternative methods for prediction of five binary traits in 23andMe.** We analyzed five binary traits, **a** any CVD, **b** depression, **c** migraine diagnosis, **d** morning person, and **e** SBMN. PRS are trained using 23andMe data that available for five populations: African American, Latino, EAS, EUR, and SAS, and then tuned in an independent set of individuals from 23andMe of the corresponding ancestry. Performance is reported based on adjusted AUC accounting for sex, age, PC1-5 in a held-out validation sample of individuals from 23andMe of the corresponding ancestry. The ratio of sample sizes for training, tuning and validation is roughly about 7:2:1, and detailed numbers are in Supplementary Data 7 and 8. The PRS-CSx package is restricted to SNPs from HM3, whereas other alternative methods use SNPs from either HM3 or MEGA. LDpred2 and its corresponding EUR and weighted methods are excluded to avoid mis-interpretation, as a result of our collaboration restrictions with 23andMe, Inc., preventing us from updating these methods to the latest version of its package. Bars in the figure show the performance of adjusted AUC for each method in each dataset. Colors are described on the right side of the figure. Source data are provided in Supplementary Data 9.

ensemble of penalized regression models to improve the performance of PRS across diverse populations. We show that PROSPER is a uniquely promising method for generating powerful PRS in multi-ancestry settings through extensive simulation studies, analysis of real datasets across a diverse type of complex traits, and considering the most recent developments of alternative methods. Computationally, the method is an order of magnitude faster compared to PRS-CSx[34], an advanced Bayesian method, and comparable to CT-SLEB[21], which derives the underlying PRS in closed forms. We have packaged the algorithm into a command line tool based on the R programming language (https://github.com/Jingning-Zhang/PROSPER).

We compare PROSPER with a number of alternative simple and advanced methods using both simulated and real datasets. The simulation results show that PROSPER generally outperforms other existing multi-ancestry methods when the target sample size is large (Fig. 2b). However when the sample size of the target population is small (Fig. 2a), no method performed uniformly the best. In this setting, when the degree of polygenicity is the lowest ($p_{causal} = 5 \times 10^{-4}$), CT-SLEB outperforms other methods by a noticeable margin, and PROSPER performs slightly worse than PRS-CSx. Simulations also show that in the scenario of a highly polygenic trait ($p_{causal} = 0.01$), irrespective of sample size, both weighted lassosum2 and PROSPER tend to exhibit

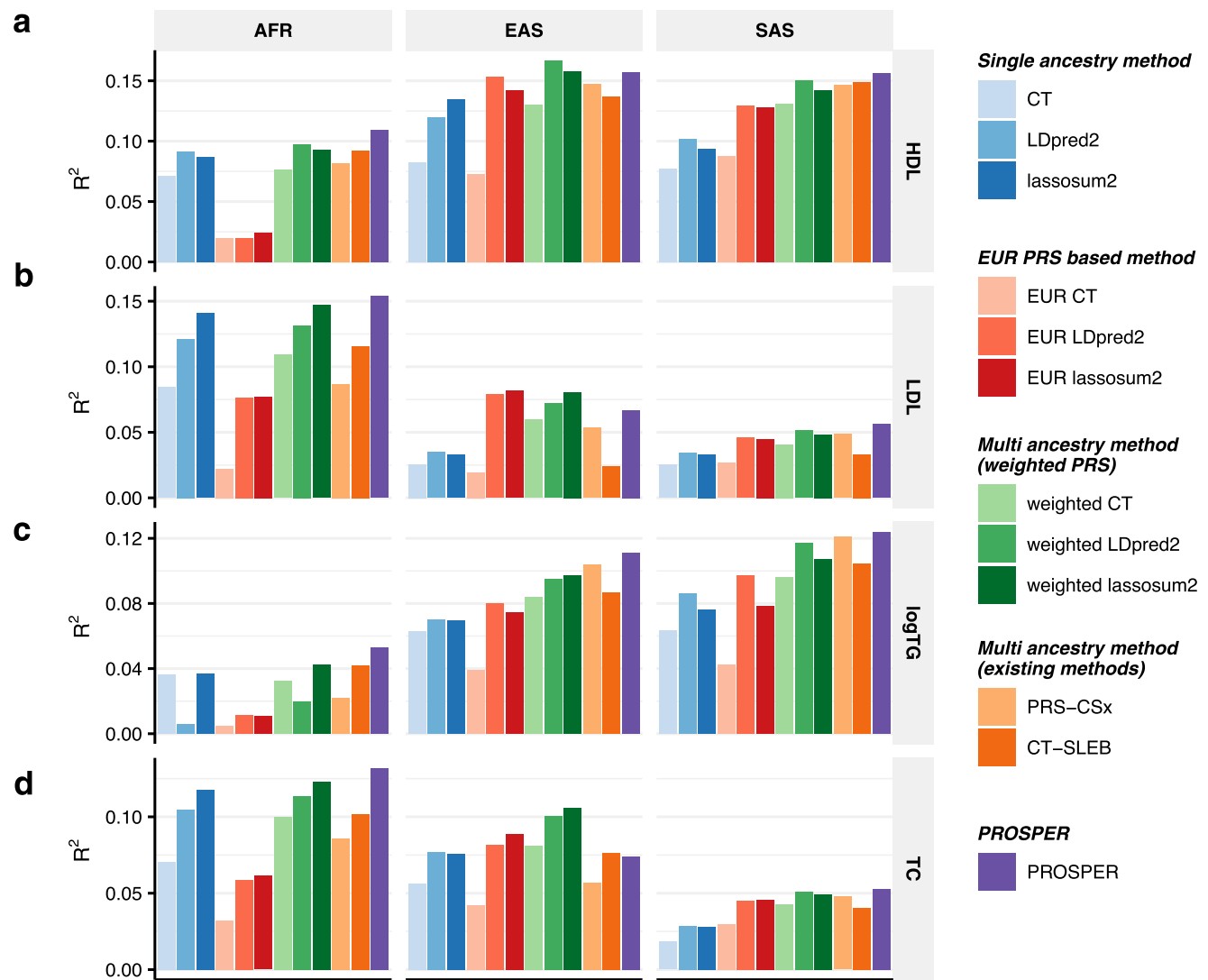

**Fig. 5 | Performance comparison of alternative methods for prediction of four blood lipid traits (GLGC-training and UKBB-tuning/validation).** We analyzed four blood lipid traits, **a** HDL, **b** LDL, **c** logTG, and **d** TC. PRS are trained using GLGC data that available for five populations: admixed African or African, East Asian, European, Hispanic, and South, and then tuned in individuals from UKBB of the corresponding ancestry: AFR, EAS, EUR, AMR, and SAS (see "Real data analysis" under "Methods" for ancestry composition). Performance is reported based on adjusted $R^2$ accounting for sex, age, PC1-10 in a held-out validation sample of individuals from UKBB of the corresponding ancestry. Sample sizes for training, tuning and validation data are in Supplementary Data 7 and 8. Results for AMR are not included due to the small sample size of genetically inferred AMR ancestry individuals in UKBB. The PRS-CSx package is restricted to SNPs from HM3, whereas other alternative methods use SNPs from either HM3 or MEGA. Bars in the figure show the performance of adjusted $R^2$ for each method in each dataset. Colors are described on the right side of the figure. Source data are provided in Supplementary Data 11.

---

superiority compared to all other methods. In terms of computational time, PROSPER is an order of magnitude faster than PRS-CSx in a five-ancestry analysis. The memory usage for PRS-CSx is smaller than PROSPER, but both are acceptable (Supplementary Data 6).

We observe that for the analysis of both continuous and binary traits using 23andMe Inc. data, PROSPER demonstrates a substantial advantage over all other methods for the AA and Latino populations, which have the largest sample sizes among all minority groups. The result is consistent with the superior performance of PROSPER observed in simulation settings when the sample size of the target population is large. However, it is worth noting that even for the two other populations, EAS and SAS, which have much smaller sample sizes, PROSPER still performs the best in half of the settings (the last two panels in Figs. 3a, b and 4a–e). For the prediction of blood lipid traits, PROSPER and weighted PRS methods perform noticeable better than other alternative methods. For the analysis of two anthropometric traits using training data from AoU, we observe that methods

that explicitly model and account for LD differences (e.g., lassosum2, LDpred2, and their corresponding weighted methods) generally achieve higher predictive accuracy than CT-based methods which discard correlated SNPs. The result is consistent with what we have observed in simulation settings under extreme polygenic architectures as expected for complex traits like height and BMI. In addition, we observe significant improvement in PRS performance using PROSPER over advanced weighted lassosum2 method which is allowed to incorporate a super learning step in lassosum2. This suggests that the additional gain of PROSPER arises from modeling shared effects across populations through the $\mathscr{L}_2$-penalty function.

PROSPER, while showing promising results in our simulations and real-data analyses, does have several limitations. First, when the sample size for the training sample for a target population is small, particularly for traits with low polygenicity, the method may not perform as well as some of the other existing methods (Fig. 2a). In this specific scenario where the number of true causal variants is small, a potential

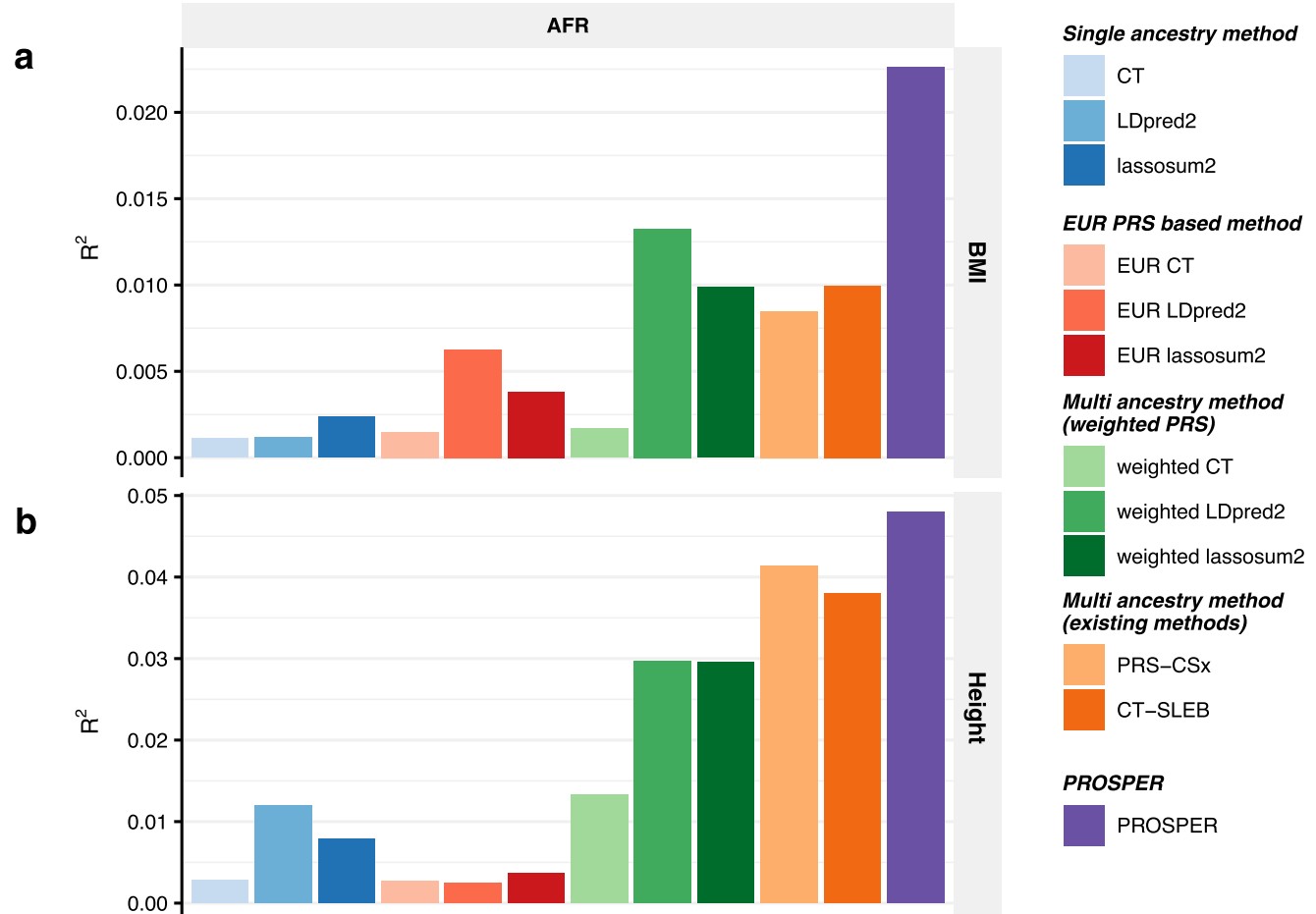

**Fig. 6 | Performance comparison of alternative methods for prediction of two anthropometric traits (AoU-training and UKBB-tuning/validation).** We analyzed two anthropometric traits, **a** BMI and **b** height. PRS are trained using AoU data that are available for three populations: African, Latino/Admixed American, and European and then tuned in individuals from UKBB of the corresponding ancestry: AFR, AMR, and EUR (see "Real data analysis" under "Methods" for ancestry composition). Performance is reported based on adjusted $R^2$ accounting for sex, age, PC1-10 in a held-out validation sample of individuals from UKBB of the corresponding ancestry. Sample sizes for training, tuning and validation data are in Supplementary Data 7 and 8. Results for AMR are not included due to the small sample size of genetically inferred AMR ancestry individuals in UKBB. The number of SNPs analyzed in AoU analyses is much smaller than other analyses because the GWAS from AoU is on array data only (see Supplementary Data 7 for the number of SNPs). The PRS-CSx package is restricted to SNPs from HM3, whereas other alternative methods use SNPs from either HM3 or MEGA. Bars in the figure show the performance of adjusted $R^2$ for each method in each dataset. Colors are described on the right side of the figure. Source data are provided in Supplementary Data 13.

reason for the suboptimal performance of PROSPER is the bias induced by lasso. This inspires future work of extending PROSPER to adaptive lasso[52] for unbiased estimation and other forms of penalty functions for sparser solutions. Second, the use of a super learning step in PROSPER can lead to poorer performance compared to weighted lassosum2 when the sample size for the tuning dataset is not adequately large. In the analysis of lipid traits for EAS, for example, we observe lower predictive accuracy of PROSPER than weighted lassosum2 (the middle panel in Fig. 5b, d). This can be attributed to overfitting in the tuning sample, as the number of tuning samples of EAS origin in the UKBB is only ~1000, while the number of PRSs combined in the super learning step is close to 500. In this scenario, we suggest comparing the performance of the ensemble PRS with that without the ensemble step, as the latter one will be more resilient to overfitting. We conducted simulation analyses to further explore the ideal sample size for tuning (Supplementary Fig. 13). Generally, a tuning sample size within the range of 1000–3000 is adequate for continuous traits. Third, we used a constant tuning parameter for the genetic similarity penalty, disregarding varying genetic distances among populations[53]. However, introducing additional tuning parameters could result in both

computational challenges and numerical instability. We have investigated this by analyzing GLGC data (see Supplementary Data 17 and "Methods"), adding an extra tuning parameter to accommodate adaptable distances between the AFR population and others. Results indicate a disproportionate increase in computational load (the last column in Supplementary Data 17) relative to the marginal enhancement in predictive accuracy, and a potential of instability and overfitting (gray cells in Supplementary Data 17). Lastly, the framework is modeled on a standardized genotype scale characterized by strong negative selection; however, there could be diverse genetic architectures in reality. To address this limitation, models could be extended to varying degrees of negative selection by multiplied by exponentiations of allele frequencies, as discussed in ref. 21.

PROSPER and a number of other recent methods have been developed for modeling summary statistics data across discrete populations typically defined by self-reported ancestry information. Increasing sample size for reference sample sizes for various populations well-matched with those providing training datasets can further enhance the performance of PROSPER and other methods that explicitly incorporates LD information into modeling. Further, there is an

emerging need to consider the underlying continuum of genetic diversity across populations in both the development and implementational of PRS in diverse populations in the future[54]. Toward this goal, a recent method called GAUDI[55] has been proposed based on the fused lasso penalty for developing PRS in admixed population using individual-level data. While GAUDI shares similarities with PROSPER in terms of the use of the lasso-penalty function, the two methods are distinct in terms of the specification of tuning parameters and use of the ensemble step. Our model specification of PROSPER makes it easily amendable to handle continuous genetic ancestry data, but further research is needed for scalable implementation of the method with individual-level data and extensive empirical evaluations.

To conclude, we have proposed PROSPER, a statistically powerful and computationally scalable method for generating multi-ancestry PRS using GWAS summary statistics and additional tuning and validation datasets across diverse populations. While no method is uniformly powerful in all settings, we show that PROSPER is the most robust among a large variety of recent methods proposed across a wide variety of settings. As individual-level data from GWAS of diverse populations becomes increasingly available, PROSPER and other methods will require additional considerations for incorporating continuous genetic ancestry information, both global and local, into the underlying modeling framework.

## Methods

We confirm that our research complies with all relevant ethical regulations. All individuals from 23andMe included have provided informed consent and answered surveys online according to our human subject protocol reviewed and approved by Ethical & Independent Review Services, a private institutional review board (http://www.eandireview.com). All participants from UK Biobank provided written informed consent (more information is available at https://www.ukbiobank.ac.uk/2018/02/gdpr/). The information of individuals from All of US included in our analyses has been collected according to All of Us Research Program Operational Protocol (https://allofus.nih.gov/sites/default/files/aou_operational_protocol_v1.7_mar_2018.pdf). The detailed consent process of All of Us is described on https://allofus.nih.gov/about/protocol/all-us-consent-process.

### Data preparation and formatting in PROSPER

We match SNPs and their alleles in GWAS summary statistics and genotypes of individuals for tuning and validation purposes to that in 1000 G reference data (phase 3)[51]. To simplify computing huge-dimensional LD matrix, we use existing LD block information from EUR[28] to divide the whole genome, and assume the blocks to be independent. We use PLINK1.9[56] with flag --r bin4 to compute the LD matrix within each block in each ancestry for common SNPs (MAF > 0.01) either in HM3[49] or the MEGA[50]. For SNPs not common in all populations, we only model them in the populations where they are common; if a SNP is population-specific that is only common in one population, we model it only using the lasso penalty without the genetic similarity penalty. The parameter path of the tuning parameter $\lambda$ for the scale factor in lasso penalty is set to a sequence evenly spaced

on a logarithmic scale from $\lambda^{\max} = \min\limits_{1 \le i \le m} \left( \dfrac{\max\limits_{1 \le k \le p}(|r_{ik}|)}{\lambda_i^0} \right)$ to

$\lambda^{\min} = 0.001 \times \lambda^{\max}$ which is set to guarantee non-zero solutions, where $r_{ik}$ is the GWAS summary statistics for the $k$-th SNP in the $i$-th population, and $\lambda_i^0$ is the underlying values of optimal tuning parameter $\lambda$ for the $i$-th population. The parameter path for the tuning parameter $c$ for the genetic similarity penalty is set to a sequence evenly spaced on a quad-root scale from $c^{\min} = 2$ to $c^{\max} = 100$, i.e., seq($c^{\min}$^(1/4), $c^{\max}$^(1/4), length.out = 10)^4 using R command. For all analyses excluding 23andMe, the length of sequences of both parameters are set to be 10,

while for the analysis of 23andMe, it is set to be 5 to reduce the computation workload caused by the confidential requirements of the 23andMe dataset.

### Obtain PROSPER solution

For $M$ populations, the objective function to minimize for $p_i$-dimensional vector of SNP effect, $\boldsymbol{\beta}_i, i = 1, \ldots, M$, is

$$\mathbf{L}(\boldsymbol{\beta}_1, \ldots, \boldsymbol{\beta}_m) = \sum_{1 \le i \le M} \left( \boldsymbol{\beta}_i^T (\mathbf{R}_i + \delta_i \mathbf{I}) \boldsymbol{\beta}_i - 2\boldsymbol{\beta}_i^T \mathbf{r}_i + 2\lambda_i \|\boldsymbol{\beta}_i\|_1^1 \right)$$
$$+ \sum_{1 \le i_1 < i_2 \le M} c_{i_1 i_2} \| \boldsymbol{\beta}_{i_1}^{s_{i_1 i_2}} - \boldsymbol{\beta}_{i_2}^{s_{i_1 i_2}} \|_2^2$$

where $\mathbf{R}_i$ is an estimate of $p_i$-by-$p_i$ LD matrix based on a reference sample from the $i$-th population, $\mathbf{r}_i$ is the $p_i$-dimensional vector of GWAS summary statistics in the $i$-th population, $\boldsymbol{\beta}_{i_1}^{s_{i_1 i_2}}$ and $\boldsymbol{\beta}_{i_2}^{s_{i_1 i_2}}$ denote the effect vectors for the SNPs shared across $i_1$-th and $i_2$-th populations (the set of SNPs is denoted by $s_{i_1 i_2}$); $\delta_i, \lambda_i$ and $c_{i_1 i_2}$ are tuning parameters as defined in above sections.

This optimization can be solved using coordinate descent algorithms by iteratively updating each element in the vectors. We take derivative for SNP $k$ in $i$-th population, $k = 1, \ldots, p_i$, $i = 1, \ldots, M$

$$\frac{\partial \mathbf{L}(\boldsymbol{\beta}_1, \ldots, \boldsymbol{\beta}_m)}{\partial \beta_{ik}} = 2 \left( 1 + \delta_i + \sum_{i' \ne i, 1 \le i' \le M} c_{ii'} \right) \beta_{ik} + 2\lambda_i \frac{\partial |\beta_{ik}|}{\partial \beta_{ik}}$$
$$- 2 \left( r_{ik} - \sum_{k' \ne k, 1 \le k' \le p} R_{i,k'k} \beta_{ik'} + \sum_{1 \le i' \le M, \text{s.t.} k \in S_{i,i'}} c_{ii'} \beta_{i'k} \right)$$

where $\beta_{ik}$ denotes the effect for SNP $k$ in $\boldsymbol{\beta}_i$, $r_{ik}$ denotes the summary statistics for SNP $k$ in $\mathbf{r}_i$, and $R_{i,k'k}$ denotes LD between the SNP $k$ and the SNP $k'$ in $\mathbf{R}_i$.

By solving $\frac{\partial \mathbf{L}(\boldsymbol{\beta}_1, \boldsymbol{\beta}_m)}{\partial \beta_{ik}} = 0$ after the $(t)$-th iteration, we can get the updating rule for the $(t+1)$-th iteration

$$\beta_{ik}^{(t+1)} = \frac{\text{sign}(u_{ik}) \cdot \max\{0, |u_{ik}| - \lambda_i\}}{1 + \delta_i + \sum\limits_{1 \le i' \le M, \text{s.t.} k \in S_{i,i'}} c_{ii'}}$$

where

$$u_{ik} = r_{ik} - \sum_{k' \ne k, 1 \le k' \le p} R_{i,k'k} \beta_{ik'}^{(t)} + \sum_{1 \le i' \le M, \text{s.t.} k \in S_{i,i'}} c_{ii'} \beta_{i'k}^{(t)}$$

### Super learning

After getting PRSs for all populations under all tuning parameter settings, we further apply super learning to combine them to be trained on the tuning samples to get the final PROSPER model and tested on the validation samples. We use the function "*SuperLearner*" implemented in the R package with the same name, and include three linear prediction algorithms: lasso, ridge, and linear regression for continuous outcomes; and two prediction algorithms: lasso and linear regression for binary outcomes. We did not include ridge for binary outcomes due to the unavailability of ridge for binary outcomes in the function. For the included algorithms which have parameters: (1) in lasso, we use 100 values in lambda path calculated in the default setting in glmnet package; (2) in ridge, we use a lambda path of sequence from 1 to 20 incrementing by 0.1. We use Area under the ROC curve (AUC) as the objective function for binary outcomes and thus use the flag "method = method. AUC" in the function.

## Existing PRS methods

We compare five groups of PRS methods. The first group is: single-ancestry method, which contains commonly known single-ancestry methods, including CT, LDpred2, and lassosum2, that are trained from the GWAS data from the target population. The second group is: EUR PRS-based method, which is the three above single-ancestry methods trained from EUR GWAS data. The third group is: weighted PRS, which uses the weights estimated from a linear regression to combine the PRSs estimated from the corresponding single-ancestry method from all populations. The fourth group is: existing multi-ancestry methods, which includes two recently published and well-performed multi-ancestry methods, PRS-CSx and CT-SLEB. The last group is our proposed PROSPER. For all algorithms that have tuning parameters or weights, the optimal ones are determined based on predictive $R^2$ or AUC on tuning samples and finally evaluated on validation samples.

Below are detailed descriptions of the existing PRS methods used as comparisons in this manuscript. In short, CT and CT-SLEB are methods that use less-dependent genetic variants after a clumping step in models. LDpred2 and PRS-CSx are Bayesian methods that can account for LD among genetic variants. Lassosum2 and our proposed PROSPER are penalized regression methods capable of modeling genome-wide genetic variants and fitting the model in a speedy way. As for the three multi-ancestry methods, CT-SLEB and PRS-CSx model the cross-ancestry genetic correlation using a multivariate Bayesian prior, while our proposed PROSPER uses a ridge penalty to impose effect-size similarity across pairs of populations.

***CT*** is implemented in our analysis by using $r^2$-cutoff of 0.1 in the clumping step and then thresholding by treating $P$ value-cutoff as a tuning parameter and being chosen from $5 \times 10^{-8}$, $1 \times 10^{-7}$, $5 \times 10^{-7}$, $1 \times 10^{-6}$,..., $5 \times 10^{-1}$, 1.0; $P$ value is from GWAS summary statistics using Chi-squared test.

***LDpred2*** is a PRS method that uses a spike-and-slab prior on GWAS summary statistics and modeling LD across SNPs. We implement LDpred2 by the function *"snp_ldpred2_grid"* in the R package "bigsnpr" version 1.12. The two tuning parameters in the algorithm include: the proportion of causal SNPs, which is chosen from a sequence of length 21 that are evenly spaced on a logarithmic scale from $10^{-5}$ to 1; per-SNP heritability, which is chosen from 0.3, 0.7, 1, or 1.4 times the total heritability estimated by LD score regression divided by the number of causal SNPs. We fix the additional "sparse" option (for truncating small effects to zero) to FALSE.

***lassosum2*** is a PRS method that uses lasso regression on GWAS summary statistics for a single ancestry. We implement lassosum2 by the function *"snp_lassosum2"* in the R package "bigsnpr" version 1.12. The two tuning parameters in the algorithm include: tuning parameter for the lasso penalty, which is chosen from a sequence of length 30 that are evenly spaced on a logarithmic scale from $0.01 \times \max_{1 \le k \le p}(|r_k|)$ to $\max_{1 \le k \le p}(|r_k|)$; and regularization parameter for LD matrix, which is chosen from c(0.001, 0.01, 0.1, 1).

***EUR PRS*** are the PRSs trained from EUR GWAS using the above single-ancestry methods, CT, LDpred2, and lassosum2, that are then applied to individuals of the target population. There is no need to perform tuning for them because the models have been tuned in EUR tuning samples. When computing scores for EUR PRS-based method, we exclude SNPs that are not presented in the validation samples from the target population.

***Weighted PRS*** linearly combines the corresponding single-ancestry method trained from all populations. The weights in the linear combination are estimated by a simple linear regression in the tuning samples from the target population.

***PRS-CSx*** is a Bayesian multi-ancestry PRS method that jointly models GWAS summary statistics and LD structures across multiple populations using a continuous shrinkage prior. It has a further step to linearly combine the posterior effect-sizes estimates for EUR and the target population using weights in a simple linear regression in the tuning samples from the target population. We implement PRS-CSx using their Python-based command line tool "PRS-CSx, available at https://github.com/getian107/PRScsx. The parameter phi was chosen from the default candidate values, $1, 10^{-2}, 10^{-4}$ and $10^{-6}$. Due to the package restriction, the models are fitted with only HM3 SNPs.

***CT-SLEB*** is a multi-ancestry PRS method that starts from clumping and thresholding, then uses Empirical-Bayes (EB) method to estimate the coefficients of PRS, and finally combines PRS by a super learning model. We implement CT-SLEB by codes available at https://github.com/andrewhaoyu/CTSLEB. The three tuning parameters in the algorithm include: $r^2$-cutoff and base size of the clumping window size used in the clumping step, which are chosen from (0.01, 0.05, 0.1, 0.2, 0.5) and (50 kb, 100 kb), respectively; and $P$ value cutoffs for EUR and the target population, which are chosen from $5 \times 10^{-8}, 5 \times 10^{-7}, 5 \times 10^{-6}, \ldots, 5 \times 10^{-1}$ and 1.0; $P$ value are from GWAS summary statistics using Chi-squared test.

## Simulation analysis

The simulated data are generated in ref. 21. In brief summary, the data were simulated under five assumed genetic architecture (as described in the legends of Fig. 2 and Supplementary Figs. 2–5) and three different degrees of polygenicity $p_{causal} = 0.01, 0.001$, and $5 \times 10^{-4}$. The sample sizes for GWAS training data are assumed to be $n = 15,000$ and $n = 80,000$ for the four non-EUR target populations; and is fixed at $n = 100,000$ for the EUR population. PRS generated from all methods are tuned in $n = 10,000$ samples, and then tested in $n = 10,000$ independent samples in each target population. We randomly repeated the simulation three times, and reported the average $R^2$ for all candidate methods.

## Computational time and memory usage

The computational time and memory usage of PROSPER and PRS-CSx are compared based on the analysis using simulated data on chromosome 22. The analysis starts from inputting all required data into the algorithms, such as summary statistics and LD reference data, and ends with outputting the final PRS coefficients from the algorithms. PROSPER requires an input of optimal parameters in single-ancestry analysis, so we also include the step of running the single-ancestry analysis, lassosum. The analyses are performed using a single core with AMD EPYC 7702 64-Core Processors running at 2.0 GHz. The reported results are averaged over 10 replicates. The sample size for training GWAS summary statistics is $n = 15,000$ for non-EUR populations and $n = 100,000$ for EUR population. The sample size for the tuning dataset is $n = 10,000$ for each population.

## Real-data analysis

Training GWAS summary statistics are from 23andMe, GLGC, and AoU. Tuning and validation of individual-level data are from 23andMe and UKBB. LD reference data are from 1000G. Detailed descriptions of those datasets are listed below.

**1000G data.** We used samples in five populations, AFR, AMR, EAS, EUR, and SAS from 1000 Genomes Project (Phase 3)[51]. The components of the five populations are described in https://useast.ensembl.org/Help/Faq?id=532.

**23andMe data.** We analyzed two continuous traits, heart metabolic disease burden and height; and five binary traits, any CVD, depression, migraine diagnosis, morning person and SBMN, using GWAS summary statistics obtained from 23andMe Inc. Data on these seven traits are available for all five populations: AA, EAS, EUR, Latino, and SAS. The LD reference panels used for the five populations, respectively, are unrelated individuals from 1000G of AFR, EAS, EUR, AMR, and SAS origins. The tuning and validation are performed on a

set of independent individuals of the corresponding ancestry from 23andMe participant cohort. Please see Supplementary Data 7 for training sample sizes and Supplementary Data 8 for tuning and validation sample sizes. The data we used are preprocessed in ref. 21, accessible from https://dataverse.harvard.edu/dataset.xhtml?persistentId=doi:10.7910/DVN/3NBNCV. The details of the data, including genotyping, quality control, imputation, removing related individuals, ancestry determination, and the preprocessing of GWAS, are described in pages 54–61 in its Supplementary Notes, and Manhattan plots and QQ plots were shown in its Supplementary Figs. 9–15. For continuous traits, we evaluate PRS performance by the predictive $R^2$ of the PRS for residualized trait values obtained from regressing the traits on covariates. For binary traits, we evaluated PRS performance by the AUC by using the roc.binary function in the R package RISCA version 1.0[57]. To compare the PRS performance for two different methods, we used the relative increase of logit-scale variance. The logit-scale variance of binary traits is converted from AUC by the formula $\sigma^2 = 2\phi^{-1}(AUC)$, where $\phi$ is the cumulative distribution function of the standard normal distribution.

**GLGC data.** We analyzed four blood lipid traits, LDL, HDL, logTG and TC, using GWAS summary statistics computed without UKBB samples that are publicly available from GLGC. Detailed information about the design of the study, genotyping, quality control, and GWAS is described in ref. 37. The data we used are preprocessed in ref. 21 in pages 61–62 in its Supplementary Notes, and Manhattan plots and QQ plots were shown in its Supplementary Figs. 16–19. Data on the four traits are available for all five populations: admixed African or African, EAS, EUR, Hispanic, and SAS. The LD reference panels used for the five populations, respectively, are unrelated individuals from 1000G of AFR, EAS, EUR, AMR, and SAS origins. The tuning and validation are performed on UKBB individuals (as described below) from the same reference ancestry label as the LD reference panel. Please see Supplementary Data 7 for sample sizes and the number of SNPs included in the analysis.

**AoU data.** We analyzed two anthropometric traits, BMI and height, using GWAS summary statistics trained from AoU. Data for the two traits are available for three ancestries: AFR, Latino/Admixed American, and EUR. The data we used are preprocessed in ref. 21, accessible from https://dataverse.harvard.edu/dataset.xhtml?persistentId=doi:10.7910/DVN/FAWEQK. The details of the data are described in pages 61–62 in its Supplementary Notes, and Manhattan plots and QQ plots were shown in its Supplementary Figs. 20 and 21. The LD reference panel used for the three populations, respectively, are 1000G unrelated individuals of AFR, AMR, and EUR origins. The tuning and validation are performed using UKBB individuals (as described below) from the same reference ancestry label as the LD reference panel. Please see Supplementary Data 7 for sample sizes and the number of SNPs included in the analysis.

**UKBB data.** We used UKBB data only for tuning and validation purposes. The four blood lipid traits and two anthropometric traits mentioned above have direct measurements in UKBB. The ancestry label of UKBB individuals is determined by genetically predicted ancestry, which are described in pages 62–63 in the Supplementary Notes of the paper from ref. 21. Tuning and validation are based on $R^2$ of the PRS regressed on the residuals of the phenotypes adjusted by sex, age and PC1-10. Please see Supplementary Data 8 for sample sizes. We note that for PRS we tested in UKBB validation samples, we use the ancestry labels in UKBB (AFR, AMR, EAS, EUR, or SAS), instead of ancestry labels in the GWAS training data, to report the $R^2$ in the Figures, "Results", and "Discussion" of this paper.

**Extra tuning parameter for varying genetic distances**

In the discussion, we investigated adding an extra tuning parameter to accommodate adaptable distances between the AFR population and others. Specifically, the pair-wise $c_{ij}$ follows the formula

$$c_{ij} = \begin{cases} r \times c & \text{if } i \text{ or } j = AFR \\ c & \text{if } i \text{ and } j \neq AFR \end{cases}$$

where $r$ and $c$ are tuning parameters; $r$ takes values from 0.5, 1, 1.5; and $c$ takes the same sequence of candidate values as described in the first paragraph of "Methods".

### Reporting summary

Further information on research design is available in the Nature Portfolio Reporting Summary linked to this article.

## Data availability

The PRSs developed for traits in GLGC and AoU will be released through the PGS Catalog (https://www.pgscatalog.org) with publication ID PGP000595 and score IDs PGS004622-PGS004686 upon publication. Simulated genotype data for 600 K subjects from five ancestries are available at https://dataverse.harvard.edu/dataset.xhtml?persistentId=doi:10.7910/DVN/COXHAP. GWAS summary-level statistics for five ancestries from GLGC are available at http://csg.sph.umich.edu/willer/public/glgc-lipids2021/results/ancestry_specific/. GWAS summary-level statistics for three ancestries from AoU are available at https://dataverse.harvard.edu/dataset.xhtml?persistentId=doi:10.7910/DVN/FAWEQK. GWAS summary statistics for the 23andMe discovery dataset could be made available through 23andMe to qualified researchers under an agreement with 23andMe that protects the privacy of the 23andMe participants. Please visit https://research.23andme.com/collaborate/#dataset-access/ for more information and to apply to access the data. GRCh37 and GRCh38 reference genome data from Phase-3 1000 Genome Project (1000G) are available at https://www.internationalgenome.org/data. Access to UKBB individual-level data can be requested from https://www.ukbiobank.ac.uk/enable-your-research/apply-for-access. Supplementary Data files and Source Data files are provided with this paper. Source data are provided with this paper.

## Code availability

All codes for data analysis, including simulation and real-data analysis, are posted through GitHub at https://github.com/Jingning-Zhang/PROSPER_analysis (ref. 58). Codes, scripts, reference data, and toy example to perform PROSPER are publicly available at https://github.com/Jingning-Zhang/PROSPER (ref. 59). The majority of our statistical analysis was performed using R 3.6.1 and R 4.0.2, and R packages: bigsnpr_1.12.2, bigstatsr_1.5.12, doMC_1.3.8, iterators_1.0.14, inline_0.3.19, RcppArmadillo_0.12.6.4.0, Rcpp_1.0.11, MASS_7.3-60, glmnet_4.1-8, Matrix_1.6-1.1, SuperLearner_2.0-28.1, gam_1.22-2, foreach_1.5.2, nnls_1.5, caret_6.0-94, lattice_0.21-8, ggplot2_3.4.3, stringr_1.5.0, readr_2.1.4, bigreadr_0.2.5, optparse_1.7.3. The authors used Python 3.8.2 for PRS-CSx. The authors used PLINK2 for computing PRS.

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

## Acknowledgements

The authors would like to thank the research participants and employees of 23andMe, Inc. for making this work possible. Full list of members in 23andMe Research Team can be found in the Supplemental Information. The authors want to thank Liz Noblin, Melissa J. Francis, and Emily Voeglein for helping with the research collaboration agreement with Harvard T.H. Chan School of Public Health, Johns Hopkins Bloomberg School of Public Health and 23andMe, Inc. The analysis utilized the Joint High-Performance Computing Exchange at Johns Hopkins Bloomberg School of Public Health. The UK Biobank data was obtained under the UK Biobank resource application 17731. This work was funded by NIH grants: R01 HG010480-01 (J. Zhang, J.J., and N.C.), K99 CA256513-01 (H.Z.), U01 HG011719 (N.C.), and K99 HG012223 (J.J.). The All of Us Research Program is supported by the National Institutes of Health, Office of the Director: Regional Medical Centers: 1 OT2 OD026549; 1 OT2 OD026554; 1 OT2 OD026557; 1 OT2 OD026556; 1 OT2 OD026550; 1 OT2 OD 026552; 1 OT2 OD026553; 1 OT2 OD026548; 1 OT2 OD026551; 1 OT2 OD026555; IAA #: AOD 16037; Federally Qualified Health Centers: HHSN 263201600085U; Data and Research Center: 5 U2C OD023196; Biobank: 1 U24 OD023121; The Participant Center: U24 OD023176; Participant Technology Systems Center: 1 U24 OD023163; Communications and Engagement: 3 OT2 OD023205; 3 OT2 OD023206; and Community Partners: 1 OT2 OD025277; 3 OT2 OD025315; 1 OT2 OD025337; 1 OT2 OD025276. In addition, the All of Us Research Program would not be possible without the partnership of its participants.

## Author contributions

J. Zhang and N.C. conceived the project. J. Zhang, J. Zhan, J.J., and H.Z. carried out all data analyses with supervision from N.C.; H.Z. created all simulated data and ran GWAS on simulated training data with the supervision from N.C.; J. Zhan, J.O., Y.J. run GWAS for training data from 23andMe Inc. with the supervision from B.L.K.; R.Z. ran GWAS on AoU-training data with the supervision from N.C. and H.Z.; J. Zhang and C.M. developed the PROSPER software; J. Zhang and N.C. drafted the manuscript, and H.Z., J.J. provided comments. All co-authors reviewed and approved the final version of the manuscript.

## Competing interests

J. Zhan, J.O., Y.J., and B.L.K. are employed by and hold stock or stock options in 23andMe, Inc. The remaining authors declare no competing interests.

## Additional information

## 23andMe Research Team

Jianan Zhan[2], Jared O'Connell[2], Yunxuan Jiang[2] & Bertram L. Koelsch[2]

