## [Peer Review File · Nature Communications]

An Ensemble Penalized Regression Method for Multi-ancestry Polygenic Risk PredictionReviewer #1 (Remarks to the Author):

Zhang and colleagues present a new method based on summary statistics, PROSPER, for multi-ancestry training and prediction. It works fairly well in both simulations and real data analyses. It builds on top of standard penalized regression models such as lassosum and adds an L2 penalty for forcing effect sizes to be similar across populations, which is a reasonable assumption to me given results from the literature. It also uses an extra step of super learning for combining all predictive models for all populations. However, I do have a few comments / questions I would like the authors to address before I could recommend this paper for publication in Nature Communications.

Main comments:

- L170-171: Is it reasonable to assume that genetic effects are similarly similar across populations (constant c). I might expect this similarity to decrease with genetic distance between populations? Any literature on this to support either claim?
- Also, on which scale should they be similar? Do you consider negative selection here?
- L208-209: The LD reference are computed from the 1000G. Have you tried using more individuals for computing the LD? E.g. based on UKBB or 23andMe (maybe not allowed for the latter one?).
- L208 & 241: AMR in 1000G is used for the LD reference of Latino ancestry; is it really the same ancestry?
- Given the very good performance of 'weighted lassosum2', I wonder whether the extra gain from PROSPER comes from the L2-penalty that shares effects across populations, or from the extra super learning step? You could also easily consider using a super learning step for lassosum2, which would be interesting to add to the comparisons, in order to answer this question.
- L447-448: This is great that the analysis code from the paper is publicly shared. However, I cannot find the code corresponding to LDpred2 analyses for example. I am a bit surprised that LDpred2 performs consistently worse than lassosum2. Maybe consider extending the grid from L651 (both parameters) to the one currently recommended in the tutorial.

Additional comments:

- L48: "polygenetic" -> polygenic
- L53: "less than ideal" -> maybe reword
- L131: "the ridge penalty is also computationally more efficient due to its continuous derivative" -> I disagree. This is maybe true in small dimensional settings, but not in ultra-high dimensions. The sparsity offered by the L1 penalty makes it possible to tremendously speed up algorithms.
- L330: "enjoy superiority" -> maybe reword
- L331: "is an order of magnitude than" -> missing word?
- L365: "the number of tuning samples of EAS origin in the UKBB is only ~ 1000 " -> I thought it was closer to 1500-2000.
- L585: "an SNP" -> a SNP
- L592: "a quad-root scale" -> I can't find anything on Google on this. Is it $\text{seq}(0.5^{0.25}, 100^{0.25}, \text{length.out} = 10)^4$? Maybe use something more common, like a simple log-scale.
- L659-660: This is not the default sequence for delta; could you explain why you modified this?

Reviewer #2 (Remarks to the Author):

In this paper, the authors proposed a framework for cross-ethnic polygenic risk prediction. The method is clearly presented, with comprehensive comparisons with other existing methods. The method is also applied to various real datasets. My comments and suggestions are as follows:

- 1) There are several competing methods such as PRS-CSx (and CT-SLEB). Could the authors give a brief description of the differences between the principle of the methodologies/assumptions etc. of these methods vs PROSPER, to facilitate understanding by the readers?
- 2) Following (1), the differences in prior assumptions/regularization strategies, could the authors provide an (intuitive) explanation on how such differences may lead to better performance of

PROSPER in certain scenarios or genetic architectures?

For example, some differences in predictive performances were described in lines 218-223. An explanation of why such differences may arise may be of interest.

3) Does the LD reference has any impact on the predictive accuracy? eg a well-matched LD reference panel may be present for Europeans, but less likely for other populations.

4) For the use of super-learner, did the authors consider using machine learning methods as well? Also, how large a tuning dataset would the authors recommend?

5) When compared with other methods, did the authors also try to build an ensemble model combining other types of methods/models, and compare such model to PROSPER?

6) Are there any possible directions to extend the current work? The authors may further discuss them. For example, any methods to address the limitations e.g. when there are only small training sets or tuning sets, and for disorders of low polygenicity.

REVIEWER COMMENTS

Reviewer #1 Comments:

Comment:

Zhang and colleagues present a new method based on summary statistics, PROSPER, for multi-ancestry training and prediction. It works fairly well in both simulations and real data analyses. It builds on top of standard penalized regression models such as lassosum and adds an L2 penalty for forcing effect sizes to be similar across populations, which is a reasonable assumption to me given results from the literature. It also uses an extra step of super learning for combining all predictive models for all populations. However, I do have a few comments / questions I would like the authors to address before I could recommend this paper for publication in Nature Communications.

Our response:

We would like to thank the reviewer for the encouraging comments. Below, we address the comments in details.

Major Comments:

Comment:

- L170-171: Is it reasonable to assume that genetic effects are similarly similar across populations (constant c). I might expect this similarity to decrease with genetic distance between populations? Any literature on this to support either claim?

Our response:

Genetically, we agree that the similarity should decrease with genetic distance¹. However, we used this assumption just for **the feasibility of computation**. Actually, other existing multi-ancestry PRS methods, such as PRS-CSx², model multiple populations in a same way, not specifically considering the genetic distances.

Our assumption is used to **simplify the computation** and **avoid overfitting** in tuning step. For example, if there are 5 populations and we let all pair-wise c_{ij} to be different, then there will be in total $\frac{5 \times 4}{2} = 10$ tuning parameters, leading to 10^{10} times of grid search, which is computationally infeasible, and causes overfitting in the tuning step surely.

We performed a sensitivity analysis allowing more flexible tuning parameters to accommodate larger genetic distance of the African (AFR) population from the others. Results in Supplementary Table 9 indicate a disproportionate increase in computational load relative to the marginal enhancement in predictive accuracy, and a potential of instability and overfitting. We have now added this discussion in lines 377-385.

Comment:

Also, on which scale should they be similar? Do you consider negative selection here?

Our response:

Our model is on strong negative selection scale (genotypes standardized by their SD), similar to all other methods, such as PRS-CS(x)³, LDpred(2)⁴, and lassosum(2)⁵. However, even the true underlying genetic model is mild or no negative selection, our model can still have a good performance (lines 209-222, Supplementary Figure 2 and Supplementary Figure 3). We have now commented on this issue in **Discussion** (lines 385-489).

Comment:

- L208-209: The LD reference are computed from the 1000G. Have you tried using more individuals for computing the LD? E.g. based on UKBB or 23andMe (maybe not allowed for the latter one?).

Our response:

Thank you for the understanding. We are not allowed to use 23andMe as LD reference. We will show an example of using UKBB as LD reference. Similar to LDpred2⁴ section 3.9, We used 10,000 individuals from the officially provided unrelated European ancestry sample list in UKBB as the LD reference for EUR. However, we note that we still used 1000G as the LD reference for the minority populations, considering the potential existence of relatedness among individuals from non-European ancestry in UKBB.

Below, we provide a real-data example for the blood lipid trait HDL. The LD reference for EUR consists of 10,000 unrelated EUR samples from UKBB, independent of the samples used in tuning and validation. LD references for the minority populations consist of samples from 1000G. The analyses and datasets are the same as those described in the manuscript, with analyses performed for all five ancestries, using GWAS data from GLGC, and tuning and validation samples from UKBB.

	R ² in validation EUR samples	R ² in validation AFR samples	R ² in validation EAS samples	R ² in validation SAS samples
1000G LD for all populations	0.203	0.110	0.157	0.156

UKB LD for EUR; 1000G LD for minority populations	0.215	0.111	0.165	0.153
---	-------	-------	-------	-------

In summary, the predictive R^2 using UKBB as LD reference for EUR improved 6% in EUR itself, and 5% for EAS; but for other minority populations, there is almost no improvement. Consider the amount of improvement, we have now provided the UKB LD reference within our PROSPER package for user accessibility.

Comment:

- L208 & 241: AMR in 1000G is used for the LD reference of Latino ancestry; is it really the same ancestry?

Our response:

The AMR in 1000G includes four sub-populations: MXL (Mexican Ancestry from Los Angeles USA), PUR (Puerto Rican from Puerto Rica), CLM (Colombian from Medellin, Colombia), and PEL (Peruvian from Lima, Peru) (see <https://useast.ensembl.org/Help/Faq?id=532>). We have now added some descriptions to lines 635-637.

Regarding line 208 (now 206), in the simulation study, the training, tuning, and validation samples are all simulated from 1000G AMR population. So their ancestry definition is consistent.

For line 241 (now 239), in the analysis of 23andMe, there may be some mismatches, but there is still an overlap in sample ancestry. To explain further, the samples labeled as 'Latino' in 23andMe are defined as 'Native American' and are mainly collected from the USA. So they share some geographic overlap with the samples in 1000G (MXL and PUR). We found that the 1000G AMR samples are the closest reference samples we could easily access, so we finally used these samples as the LD reference for Latino in the 23andMe data analysis. The results from Figure 3 and Figure 4 shows that the R^2 and AUC for Latino are on a similar scale as those for the other well-defined ancestries, which indirectly illustrates that the LD reference panel is feasible. However, we want to note that if there are samples that could be ancestrally better matched to the definition of Latino in 23andMe, the prediction results are expected to improve further.

We have now added a sentence to the Discussion section in lines 393-395.

Comment:

- Given the very good performance of 'weighted lassosum2', I wonder whether the extra gain from PROSPER comes from the L2-penalty that shares effects across populations, or from the extra super learning step? You could also easily consider using a super learning step for

lassosum2, which would be interesting to add to the comparisons, in order to answer this question.

Our response:

Thank you for the comment. Based on your suggestion, we have now added this comparison in lines 291-297, Supplementary Figure 6 and Supplementary Table 6.3.

Based on the results, we found that the additional gain for EAS and SAS seems to result from the joint modeling step in PROSPER, whereas for AFR, the super learning step in lassosum2 has already yielded significant improvement. This aligns with our intuition that AFR is more genetically distinct from other populations.

Comment:

- L447-448: This is great that the analysis code from the paper is publicly shared. However, I cannot find the code corresponding to LDpred2 analyses for example. I am a bit surprised that LDpred2 performs consistently worse than lassosum2. Maybe consider extending the grid from L651 (both parameters) to the one currently recommended in the tutorial.

Our response:

We apologize for not providing the directory for codes for all methods. For methods other than lassosum2, we relied on method comparison results from a previous manuscript ⁶, and the corresponding codes can be found on GitHub at https://github.com/andrewhaoyu/multi_ethnic/tree/master.

To be more specific, the codes for ldpred2, EUR ldpred2, weighted ldpred2 are available at https://github.com/andrewhaoyu/multi_ethnic/tree/master/code/Jin. Codes for CT, EUR CT, weighted CT, PRS-CSx, and CT-SLEB can be found in directories corresponding to their respective datasets. For example, the codes for analyzing GLGC are available at https://github.com/andrewhaoyu/multi_ethnic/tree/master/code/GLGC_analysis

We have now updated our main page for data analysis on GitHub (https://github.com/Jingning-Zhang/PROSPER_analysis/blob/main/README.md), as well as the Software and Code section in nr-reporting-summary.pdf.

To explain the results of ldpred2 vs lassosum2, there are actually some practical issues with ldpred2. We observed that when applying ldpred2 to GWAS with large sample sizes, the LDpred2-grid function within the bigsnpr package returns NA for certain tuning parameter settings, for reasons that are currently unknown. We suspect that one potential reason might be the slightly different LD patterns between the training GWAS and the reference panel, which might influence convergence, particularly when the training GWAS is large.

In our simulation study, the simulated EUR GWAS was fixed and had a large sample size of 100K. So in some tuning parameter settings, LDpred2-grid produced NA output. This problem was extremely severe in 23andMe data analysis where all tuning parameter settings resulted in NA output from the function of LDpred2-grid. This was why in the 23andMe data analysis, we had to use LDpred2-auto function instead, which does not perform grid search, but estimates the parameters automatically. The prediction accuracy of PRS from LDpred2-auto is usually worse than that from LDpred2-grid. Therefore, ldpred2 performed much worse in 23andMe data analysis. In AoU data analysis, where the GWAS sample size for all populations are not as large, ldpred2 and lassosum2 are comparable -- lassosum2 performs better for BMI, while ldpred2 performs better for height. Thus, we will use AoU as an example below to show the results using the default tuning parameter settings currently recommended in the LDpred2 tutorial.

As our main focus is not on single-ancestry Bayesian methods, we didn't discuss this issue in the manuscript. Our primary goal is to compare with other multi-ancestry PRS methods.

The table shows the predictive R^2 in validation AFR samples for models trained using AoU data.

	BMI	Height
Ldpred2 with the same sequence in the manuscript	0.0005	0.0070
Ldpred2 with the recommended sequence in tutorial	0.0009	0.0089
Lassosum2	0.0021	0.0044

Based on these results, using the recommended sequence in tutorial indeed increases the predictive R^2 but we still arrive at the same conclusion – lassosum2 performs better for BMI; and ldpred2 performs better for height.

Additional comments:

- L48: “polygenetic” -> polygenic

Our response: We have modified the text.

- L53: “less than ideal” -> maybe reword

Our response: We have modified the text.

- L131: “the ridge penalty is also computationally more efficient due to its continuous derivative” -> I disagree. This is maybe true in small dimensional settings, but not in ultra-high dimensions. The sparsity offered by the L1 penalty makes it possible to tremendously speed up algorithms.

Our response: Thank you for the comment. Your reasoning makes a lot sense. We have now removed that sentence.

- L330: “enjoy superiority” -> maybe reword

Our response: We have modified the text.

- L331: “is an order of magnitude than” -> missing word?

Our response: We have modified the text.

- L365: “the number of tuning samples of EAS origin in the UKBB is only ~1000” -> I thought it was closer to 1500-2000.

Our response: We half-half split the dataset into tuning and testing. So the number of tuning sample is less than 1000.

- L585: “an SNP” -> a SNP

Our response: We have modified the text.

- L592: “a quad-root scale” -> I can’t find anything on Google on this. Is it ‘seq(0.5^0.25, 100^0.25, length.out = 10)^4’? Maybe use something more common, like a simple log-scale.

Our response: Sorry for the confusion. Yes it is.

The log-scale is too loose at the beginning, and may miss the optimal value. We have now added an explanation to the formula.

A comparison of the quad-root and log scale:

```
> seq(0.5^0.25, 100^0.25, length.out = 10)^4
```

```
[1] 0.500000 1.457868 3.388525 6.797619 12.297022 20.604833
```

```
[7] 32.545375 49.049197 71.153073 100.000000
```

```
> exp(log(seq(0.5, 100, length.out = 10)))
```

```
[1] 0.500000 11.55556 22.61111 33.66667 44.72222 55.77778 66.83333
```

```
[8] 77.88889 88.94444 100.00000
```

- L659-660: This is not the default sequence for delta; could you explain why you modified this?

Our response: We first tried the default sequence but found that sometimes the optimal delta can reach the top boundary of that sequence. So we gradually increase the range of the sequence and finally got a sequence that could probably cover all scenarios, which is the one in the manuscript.

Reviewer #2 Comments:

Comment:

In this paper, the authors proposed a framework for cross-ethnic polygenic risk prediction. The method is clearly presented, with comprehensive comparisons with other existing methods. The method is also applied to various real datasets. My comments and suggestions are as follows:

Our Response:

We would like to thank the reviewer for all the encouraging comments and providing a detailed scrutiny of the paper.

Major Comments:

Comment:

1) There are several competing methods such as PRS-CSx (and CT-SLEB). Could the authors give a brief description of the differences between the principle of the methodologies/assumptions etc. of these methods vs PROSPER, to facilitate understanding by the readers?

Our response:

Thank you for the suggestion. We have added a paragraph in lines 566-573, briefly summarizing and comparing different methods.

Comment:

2) Following (1), the differences in prior assumptions/regularization strategies, could the authors provide an (intuitive) explanation on how such differences may lead to better performance of PROSPER in certain scenarios or genetic architectures?
For example, some differences in predictive performances were described in lines 218-223. An explanation of why such differences may arise may be of interest.

Our response:

We have added some discussion in lines 364-367 and 385-389. In simulations involving low polygenicity and a small training sample size, lasso could introduce bias that may have a more pronounced impact due to the limited number of true causal variants and the noise in the training data. In addition, our framework is structured based on a standardized genotype scale

characterized by strong negative selection. Therefore, as the strength of negative selection increase, our PROSPER might potentially suffer less from such bias.

Comment:

3) Does the LD reference has any impact on the predictive accuracy? eg a well-matched LD reference panel may be present for Europeans, but less likely for other populations.

Our response:

In response to the reviewer #1, we demonstrate the comparison of using UKBB as LD reference for EUR.

Below, we provide a real-data example for the blood lipid trait HDL. The LD reference for EUR consists of 10,000 unrelated EUR samples from UKBB, independent of the samples used in tuning and validation. LD references for the minority populations consist of samples from 1000G. The analyses and datasets are the same as those described in the manuscript, with analyses performed for all five ancestries, using GWAS data from GLGC, and tuning and validation samples from UKBB.

	R ² in validation EUR samples	R ² in validation AFR samples	R ² in validation EAS samples	R ² in validation SAS samples
1000G LD for all populations	0.203	0.110	0.157	0.156
UKB LD for EUR; 1000G LD for minority populations	0.215	0.111	0.165	0.153

In summary, the predictive R² using UKBB as LD reference for EUR improved 6% in EUR itself, and 5% for EAS; but for other minority populations, there is almost no improvement. Consider the amount of improvement, we have now provided the UKB LD reference within our PROSPER package for user accessibility.

Comment:

4) For the use of super-learner, did the authors consider using machine learning methods as well?

Our response:

For genetic prediction using common variants, there has been very little evidence of gene-gene interactions and developing additive score is usually considered adequate⁷. Therefore, we recommend to use only linear methods in super learning, such as lasso and ridge. However, we

could definitely explore more advanced machine learning methods, such as neural network and random forests.

Please find below a summary table for the blood lipid trait HDL in GLGC as a quick example for comparing super learning with and without neural network and random forests. According to these results, there is only marginal improvement in HDL prediction by adding neural network and random forest to the super learning step.

	R ² in validation AFR samples	R ² in validation EAS samples	R ² in validation SAS samples
PROSPER (super learning methods include lasso, ridge, and linear regression)	0.1096	0.1572	0.1564
PROSPER (super learning methods include lasso, ridge, linear regression, neural network, and random forest)	0.1096	0.1421	0.1613

Comment:

Also, how large a tuning dataset would the authors recommend?

Our response:

For the size of the tuning dataset, we have now added a paragraph in Discussion section (lines 375-377). Based on our simulation results, a tuning sample size within the range of 1000-3000 is generally adequate for continuous traits (Supplementary Figure 7).

Comment:

5) When compared with other methods, did the authors also try to build an ensemble model combining other types of methods/models, and compare such model to PROSPER?

Our response:

Statistically, we could perform super learning for every method with tuning parameters. However, in this study, we compared the methods as they were originally proposed and did not modify the methods themselves.

Single-ancestry methods (ldpred2, lassosum2) typically do not include an ensemble step. So we have now added Supplementary Figure 6 and Supplementary Table 6.3 as a quick example to compare the addition of super learning to lassosum2.

Multi-ancestry methods generally include an ensemble step. For weighted ldpred2 and weighted lassosum2, the weighted score itself serves as an ensemble model. For PRS-CSx, its final step involves a weighted sum of the optimal PRS across all populations, which is also an ensemble model. For CT-SLEB, the last step is exactly super learning so is also an ensemble model.

Comment:

6) Are there any possible directions to extend the current work? The authors may further discuss them. For example, any methods to address the limitations e.g. when there are only small training sets or tuning sets, and for disorders of low polygenicity.

Our response:

We proposed future works related to the continuum of genetic diversity across populations in lines 393-395. Based on your suggestion, we have now added more discussion in lines 373-375, 387-389.

Reviewer #1 (Remarks to the Author):

I thank the authors for their response. However, I still have several concerns.

- I suggested using the super learning step for 'weighted lassosum2', not just 'lassosum2' (i.e. combining all parameters + all ancestries at once, as done in PROSPER); this would be the closest method to PROSPER, and would enable the readers to really understand whether the extra L2-regularization in PROSPER provides better predictive performance. I would also like this comparison to appear in all figures, not just for 4 traits in one supplementary figure.
- I don't think it is okay to directly reuse results from another paper; I would ask the authors to rerun the methods to ensure relevant comparisons.
- The authors report severe convergence issues of LDpred2-grid (the NA values for all models). Did the authors contact the authors of LDpred2 about this to understand why there were such severe issues? When quickly looking at their code (e.g. [https://github.com/andrewhaoyu/multi_ethnic/blob/master/code/Jin/Data Analysis/glgc-EUR_LDpred2.R#L29-L31](https://github.com/andrewhaoyu/multi_ethnic/blob/master/code/Jin/Data%20Analysis/glgc-EUR_LDpred2.R#L29-L31)), I can see that they used `h2_seq <- c(0.7, 1, 1.4)` instead of this times the heritability estimate from LD score regression. I would suggest the authors to go over their code once more with an external person to verify that there is no mistake in their code.
- Similarly, when looking at the LDpred2-auto code (e.g. [https://github.com/andrewhaoyu/multi_ethnic/blob/master/code/Jin/Data Analysis/ldpred2-auto-23andme.R#L92-L96](https://github.com/andrewhaoyu/multi_ethnic/blob/master/code/Jin/Data%20Analysis/ldpred2-auto-23andme.R#L92-L96)), I do not see any initial QC of the GWAS summary statistics (maybe performed in another file?), I do not see the robust parameters suggested in the tutorial, nor the post-QC of chains from the tutorial as well. This does not seem to be a fair comparison anymore.
- For lassosum2, there is no way you need to add a delta value of 100 to the diagonal of the LD matrix. Usually 1 or 3 is already more than enough when there are some strong misspecifications. Also, it seems the code is using 0.5 as minimum value for this regularization parameter (e.g. https://github.com/Jingning-Zhang/PROSPER_analysis/blob/main/real_data_analysis/allofus/lassosum2/2_run_lassosum2.R#L29) whereas the paper says 0.01 as minimum. The need to use such strong regularization is for me a sign that there is something else wrong in the input data or in the code. I suggest the authors check the quality of the GWAS summary statistics, as well as their code.
- The sequence on a log-scale is `exp(seq(log(0.5), log(100), length.out = 10))`, not `exp(log(seq(0.5, 100, length.out = 10)))` which is just a linear scale.
- Why is this a problem to derive an unrelated set of non-European individuals in the UKBB? There are at least two ways of doing this.

Reviewer #2 (Remarks to the Author):

The authors have rather comprehensively addressed my comments or suggestions. Most of the points I raised are also discussed in the manuscript with suitable additional analyses included. The paper is well-written and I have no further major comments.

REVIEWER COMMENTS

Reviewer #1 (Remarks to the Author):

Your comment:

I thank the authors for their response. However, I still have several concerns.

Our response: Thank you for your valuable comments. In this revision, we have re-run and updated all analyses using the latest versions of packages. This took several months and we apologize for the long revision time. The results following this revision make more sense, and we hope you find satisfaction with our revisions.

Your comment:

I suggested using the super learning step for ‘weighted lassosum2’, not just ‘lassosum2’ (i.e. combining all parameters + all ancestries at once, as done in PROSPER); this would be the closest method to PROSPER, and would enable the readers to really understand whether the extra L2-regularization in PROSPER provides better predictive performance. I would also like this comparison to appear in all figures, not just for 4 traits in one supplementary figure.

Our response:

Thank you for your valuable comment. We have now incorporated your suggested method for performance comparison into all simulated data scenarios, GLGC, and the All of US (AoU) analysis.

Your suggested way of super-learning-weighting lassosum2 has been termed "advanced weighted lassosum2", denoting a super-learning combined lassosum2 PRS across all tuning parameters and ancestries. The complete results can be found in Supplementary Figures 6-10 for all simulated data scenarios, Supplementary Figure 11 for GLGC, and Supplementary Figure 12 for the All of US analysis.

Our discussion of the results is in lines 317-330 and 372-374. Briefly, PROSPER consistently has more advantage than the advanced weighted lassosum2 in simulations. In real data, their performance depends on traits and ancestries, and PROSPER has 41.1% relative improvement in R² over advanced weighted lassosum2 on average across all ancestries and all traits in GLGC and AoU. Unfortunately, we were not able to perform this analysis in 23andMe due to time constraint of the 23andMe team. We have noted this limitation in lines 329-330.

We have deposited our codes for this analysis in https://github.com/Jingning-Zhang/PROSPER_analysis/blob/main/revision2/results_from_latest_bigsnp_r_package/lassosum2_most_recent_version/Simulation/4_weighted_sl.R, https://github.com/Jingning-Zhang/PROSPER_analysis/blob/main/revision2/results_from_latest_bigsnp_r_package/lassosum2_most_recent_version/GLGC/4_weighted_sl.R, and https://github.com/Jingning-Zhang/PROSPER_analysis/blob/main/revision2/results_from_latest_bigsnp_r_package/lassosum2_most_recent_version/AoU/4_weighted_sl.R.

[Zhang/PROSPER_analysis/blob/main/revision2/results_from_latest_bigsnp_package/lassosum2_most_recent_version/AoU/4_weighted_sl.R](https://github.com/Zhang/PROSPER_analysis/blob/main/revision2/results_from_latest_bigsnp_package/lassosum2_most_recent_version/AoU/4_weighted_sl.R) .

Your comment:

I don't think it is okay to directly reuse results from another paper; I would ask the authors to rerun the methods to ensure relevant comparisons.

Our response:

Thank you for raising the concern. In fact, we conducted these projects together, using the exact same data and pipeline throughout, ensuring fair comparisons. To prevent content overlap, we didn't include the parts of the data description, data quality control, and alternative methods (including CT, ldpred2, PRS-CSx, and CT-SLEB) in our manuscript, which have already been published in Zhang et al. (2023, Nature Genetics). We have now cited this paper in line 195, 630, 662, 670-671, 689-690, 695, 701-702, and 706 hoping to make it clearer to our audience. We greatly appreciate your advice, and to enhance clarity for our audience, we have now copied the relevant codes into the same GitHub repository. Please review them in the https://github.com/Jingning-Zhang/PROSPER_analysis/tree/main/revision2/codes_related_methods_Zhang.NG.2023 .

Your comment:

The authors report severe convergence issues of LDpred2-grid (the NA values for all models). Did the authors contact the authors of LDpred2 about this to understand why there were such severe issues? When quickly looking at their code (e.g. https://github.com/andrewhaoyu/multi_ethnic/blob/master/code/Jin/Data_Analysis/glgc-EUR_LDpred2.R#L29-L31), I can see that they used `h2_seq <- c(0.7, 1, 1.4)` instead of this times the heritability estimate from LD score regression. I would suggest the authors to go over their code once more with an external person to verify that there is no mistake in their code.

Our response:

The LDpred2 analysis was conducted using version 1.8 of bigsnpr package and the corresponding tutorial available at the time of our initial analysis, which had convergence issues with our data. We have since addressed this by updating the results using the latest version, bigsnpr 1.12, and now the convergence problem has been resolved. Therefore, we have now incorporated the updated LDpred2 results using the latest bigsnpr package in our manuscript for all the analysis of simulated data, GLGC data, and All of US data. However, due to the restriction of our collaboration with 23andMe, Inc., we are currently unable to run new analyses using the 23andMe dataset. To prevent potential misinterpretation of result comparisons, we have excluded LDpred2 and its related methods from the 23andMe analysis.

The new results of LDpred2 improved over the old ones. In simulation analyses, LDpred2 and its related methods (EUR LDpred2 and weighted LDpred2) now become the best-performing one in the three candidate methods (CT, LDpred2 and lassosum2) in their corresponding categories. Considering all methods compared, weighted LDpred2 even has comparable performance with PROSPER. The results are expected given that in our simulation studies data were simulated under the spike-and-slab model, which is the assumed prior distribution for LDpred methodology. In real data analyses, the performance of LDpred2 and its related methods depends on traits and ancestries. It could outperform lassosum2 in some scenarios. We commented those changes caused by the version update of LDpred2 in lines 216, 224, 244-251, 292-293, and 489-490. In general, our conclusion remains the same.

Regarding your question about “`h2_seq <- c(0.7, 1, 1.4)`”, this is actually a column name we designated in the results table to record the tuning parameters. The actual h^2 parameter used in our analysis are the values of this sequence, i.e., `c(0.7, 1, 1.4)`, multiplied by the estimated heritability obtained from LD score regression.

To facilitate transparency and accessibility, we have deposited our codes for the latest version of bigsnpr in the https://github.com/Jingning-Zhang/PROSPER_analysis/tree/main/revision2/results_from_latest_bigsnpr_package.

Your comment:

Similarly, when looking at the LDpred2-auto code (e.g. https://github.com/andrewhaoyu/multi_ethnic/blob/master/code/Jin/Data_Analysis/ldpred2-auto-23andme.R#L92-L96), I do not see any initial QC of the GWAS summary statistics (maybe performed in another file?), I do not see the robust parameters suggested in the tutorial, nor the post-QC of chains from the tutorial as well. This does not seem to be a fair comparison anymore.

Our response:

The initial quality control for all analyzed GWAS data is detailed in Zhang et al. (2023, Nature Genetics) [ref 22 in manuscript]. The entire analysis was conducted together using the exact same data and pipeline throughout. To prevent content overlap, the initial QC for the 23andMe data analysis were incorporated into the aforementioned paper, and we have cited that paper and provided those details in lines 668-671 to provide clarity to our audience.

As mentioned in our response to your previous comment, the previous LDpred2 results, including LDpred2-auto in the 23andMe analysis, indeed had convergence issues. Again, thank you for identifying the huge problem and raising the concern. Unfortunately, due to the restriction on time of 23andMe Inc collaborators, we are unable to perform major new analyses on their dataset. To prevent any potential misinterpretation of result comparisons, LDpred2 and its related methods have been excluded from the 23andMe analysis.

Your comment:

For lassosum2, there is no way you need to add a delta value of 100 to the diagonal of the LD matrix. Usually 1 or 3 is already more than enough when there are some strong misspecifications. Also, it seems the code is using 0.5 as minimum value for this regularization parameter (e.g. https://github.com/Jingning-Zhang/PROSPER_analysis/blob/main/real_data_analysis/allofus/lassosum2/2_run_lassosum2.R#L29) whereas the paper says 0.01 as minimum. The need to use such strong regularization is for me a sign that there is something else wrong in the input data or in the code. I suggest the authors check the quality of the GWAS summary statistics, as well as their code.

Our response:

Thank you for bringing up the issue. We have now used the latest bigsnpr package (version 1.12) and strictly adhered to the most recent tutorial. The updated results have been updated to all figures and tables. Fortunately, there is almost no change in the lassosum2 results, ensuring that the conclusions remain consistent.

We have deposited our codes for this analysis in https://github.com/Jingning-Zhang/PROSPER_analysis/tree/main/revision2/results_from_latest_bigsnpr_package/lassosum2_most_recent_version.

Your comment:

The sequence on a log-scale is ``exp(seq(log(0.5), log(100), length.out = 10))``, not ``exp(log(seq(0.5, 100, length.out = 10)))`` which is just a linear scale.

Our response:

Thank you for pointing out the typo. We have corrected this error and updated it to the values suggested in the latest tutorial. The revision has been made in lines 604-607.

Your comment:

Why is this a problem to derive an unrelated set of non-European individuals in the UKBB? There are at least two ways of doing this.

Our response:

We have now used the relationship-based pruning method in plink (`--rel-cutoff`, alias: `--grm-cutoff`), aiming to maximize the final sample size by retaining unrelated individuals in the UKBB. Subsequently, we identified the unrelated sets for AFR (sample size 3049), EAS (sample size 636), and SAS (sample size 1831), and created LD panels for these ancestries. However, due

to the limited number of independent samples in UKBB for AMR (only 32), we have chosen not to generate an LD panel based on such a small set of independent samples. The LD panel based on UKBB has been released in <https://github.com/Jingning-Zhang/PROSPER>.

We further checked if the UKB LD reference performs reasonably. Here is an example in GLGC data analysis. The R^2 in testing samples are reported below. The UKB LD reference panel performs slightly better compared to the 1000G LD reference panel.

UKB LD reference (EUR,AFR,EAS,SAS from UKB; AMR from 1000G)				
	HDL	LDL	logTG	TC
AFR	0.1017	0.1606	0.0540	0.1347
EAS	0.1517	0.0672	0.1072	0.0774
SAS	0.1709	0.0584	0.1279	0.0558
1000G LD reference (EUR,AFR,AMR,EAS,SAS from 1000G)				
	HDL	LDL	logTG	TC
AFR	0.1096	0.1538	0.0528	0.1320
EAS	0.1572	0.0668	0.1110	0.0740
SAS	0.1564	0.0561	0.1238	0.0527

We have deposited our codes for this analysis in https://github.com/Jingning-Zhang/PROSPER_analysis/tree/main/revision2/UKBB_LDref.

Reviewer #2 (Remarks to the Author):

Your comment:

The authors have rather comprehensively addressed my comments or suggestions. Most of the points I raised are also discussed in the manuscript with suitable additional analyses included. The paper is well-written and I have no further major comments.

Our response: Thank you for your satisfaction with our revisions.

Reviewer #1 (Remarks to the Author):

I thank the authors for the work they put in this second revision.
I do not have any further comment.

Reviewer #2 (Remarks to the Author):

The authors have addressed most concerns raised by the reviewers and performed additional analyses. I personally have no major comments.

REVIEWER COMMENTS

Reviewer #1 (Remarks to the Author):

I thank the authors for the work they put in this second revision. I do not have any further comment.

Reviewer #2 (Remarks to the Author):

The authors have addressed most concerns raised by the reviewers and performed additional analyses. I personally have no major comments.

OUR RESPONSE TO BOTH REVIEWERS

Thank you for your satisfaction with our revisions. Again, we appreciate all your valuable comments that helped us improve the manuscript.